# Constrained Flow Optimization via Sequential Fine-Tuning for Molecular Design

Sven Gutjahr [* 1]   Riccardo De Santi [* 1 2 3]   Luca Schaufelberger [* 3 4]   Kjell Jorner [3 4]   Andreas Krause [1 2 3]

## Abstract

Adapting generative foundation models, in particular diffusion and flow models, to optimize given reward functions (e.g., binding affinity) while satisfying constraints (e.g., molecular synthesizability) is fundamental for their adoption in real-world scientific discovery applications such as molecular design or protein engineering. While recent works have introduced scalable methods for reward-guided fine-tuning of such models via reinforcement learning and control schemes, it remains an open problem how to algorithmically trade-off reward maximization and constraint satisfaction in a reliable and predictable manner. Motivated by this challenge, we first present a rigorous framework for *Constrained Generative Optimization*, which brings an optimization viewpoint to the introduced adaptation problem and retrieves the relevant task of constrained generation as a sub-case. Then, we introduce **C**onstrained **F**low **O**ptimization (CFO), an algorithm that automatically and provably balances reward maximization and constraint satisfaction by reducing the original problem to sequential fine-tuning via established, scalable methods. We provide convergence guarantees for constrained generative optimization and constrained generation via CFO. Ultimately, we present an experimental evaluation of CFO on both synthetic, yet illustrative, settings, and a molecular design task. Across these evaluations, CFO achieves consistent increases in reward while ensuring high constraint satisfaction, showcasing its practical utility for constrained generative optimization.

---

[*]Equal contribution  [1]Department of Computer Science, ETH Zurich [2]ETH AI Center [3]NCCR Catalysis, Switzerland [4]Institute of Chemical and Bioengineering, Department of Chemistry and Applied Biosciences, ETH Zurich. Correspondence to: Sven Gutjahr, Riccardo De Santi, Luca Schaufelberger <sgutjahr, rdesanti, schaluca@ethz.ch>.

*Proceedings of the 43$^{rd}$ International Conference on Machine Learning*, Seoul, South Korea. PMLR 306, 2026. Copyright 2026 by the author(s).

## 1. Introduction

Recent advances in generative modeling, particularly the advent of diffusion (Ho et al., 2020; Song et al., 2021b;a) and flow models (Lipman et al., 2023), have led to state-of-the-art performance in several fields such as image synthesis (Rombach et al., 2022), biology (Corso et al., 2023; Wohlwend et al., 2025), and chemistry (Hoogeboom et al., 2022). In particular, they have been applied for the design of protein structures (Wu et al., 2024), drug-like molecules (Dunn & Koes, 2024), and DNA sequences (Stärk et al., 2024), among others. These generative models excel at capturing complex data distributions and generating realistic samples. However, approximately sampling from the data distribution is insufficient for most real-world discovery applications, where one typically wishes to generate candidates maximizing task-specific *rewards*, a problem recently denoted by *generative optimization* (De Santi et al., 2025b; Li et al., 2024). Examples of rewards of interest include binding affinity in drug discovery (Pantsar & Poso, 2018), or drug-likeness (Bickerton et al., 2012). To tackle the generative optimization problem, recent works have introduced scalable fine-tuning methods that adapt a pre-trained flow or diffusion model to maximize a given reward function under KL-regularization from the pre-trained model, via reinforcement learning (RL) or control theoretic methods (e.g., Domingo-Enrich et al., 2025; Uehara et al., 2024b; Tang & Zhou, 2024).

**The importance of known constraints in generative optimization.** Many generative design and scientific discovery problems require generated samples to satisfy explicit, domain-specific constraints, e.g., bounded toxicity (Amorim et al., 2024), synthetic accessibility (Ertl & Schuffenhauer, 2009; Neeser et al., 2024), or biophysical plausibility of docking poses (Buttenschoen et al., 2024). Even though current fine-tuning schemes regularize toward a pre-trained model (Domingo-Enrich et al., 2025; Uehara et al., 2024b; Tang & Zhou, 2024), which limits the distributional drift, they cannot certify hard constraints to be satisfied (Uehara et al., 2024a). This limitation arises because task-specific constraints may not be encoded in the original dataset or may be learned only imperfectly from finite training data. A naive approach to address such explicit constraints would be to include them as rewards, i.e., as another term in a manually weighted objective function. However, this approach is

unreliable in practice, as the appropriate weighting between rewards and constraints varies across tasks and training phases, and needs to be determined through inefficient trial and error. Furthermore, as optimization explores high-reward regions, the chosen weights can unexpectedly favor reward at the expense of constraint satisfaction, yielding samples with attractive rewards, which, however, violate the domain-specific constraints. Driven by these limitations of current flow adaptation methods for constraint satisfaction, we pose the following question:

*How can we fine-tune a pre-trained flow or diffusion model to reliably and predictably trade-off reward optimization and constraint satisfaction?*

**Our approach.** A growing body of work demonstrates that classical optimization ideas can be meaningfully adapted to the fine-tuning of flow and diffusion models, including formulations motivated by mirror descent (Nemirovskij & Yudin, 1983; De Santi et al., 2025b), chance constraints (Ben-Tal & Nemirovski, 2000; Zhang et al., 2026), and bilevel optimization (Bracken & McGill, 1973; Xiao et al., 2025). Analogously, in this work, we aim to tackle this question by introducing a formal framework for *Constrained Generative Optimization* (Sec. 3) via flow model fine-tuning, which entails adapting a pre-trained flow model to generate samples maximizing a reward function while satisfying arbitrary constraints. Moreover, the proposed formulation retrieves the relevant task of constrained generative modeling as the sub-case where the reward function is constant. Next, we introduce **C**onstrained **F**low **O**ptimization (CFO), a dual approach based on the augmented Lagrangian scheme (Birgin & Martínez, 2014) that turns the constrained objective into a sequence of ordinary generative optimization subproblems. At a high level, CFO alternates between two steps: solving a KL-regularized fine-tuning problem (Domingo-Enrich et al., 2025; Uehara et al., 2024b) to maximize an augmented reward function, and updating the parameters of the augmented reward using estimated constraint violations on generated samples (see Sec. 4). This sequentially tunes the penalty on constraint violations, thereby avoiding the need to manually trade-off weight selection. CFO renders it possible to adapt a pre-trained flow model to maximize expected rewards while enforcing satisfaction of arbitrary constraints and preserving closeness to the pre-trained model. We provide guarantees that ensure constraint satisfaction under the realistic assumptions of an approximate solver, and that achieve reward maximization under a more idealized setting (Sec. 5). Finally, we evaluate CFO for both constrained generative optimization and modeling problems, showcasing its performance in both visually interpretable settings and in molecular design tasks, showing constrained optimization of quantum mechanical properties (Sec. 6).

**Our contributions.** We present the following contributions:
- We formulate *Constrained Generative Optimization* via flow fine-tuning, capturing the practically relevant task of reward-guided adaptation under given constraints (Sec. 3).
- We introduce **C**onstrained **F**low **O**ptimization (CFO), an augmented Lagrangian-based method that provably tackles the introduced problem via sequential fine-tuning (Sec. 4).
- We provide guarantees for constrained generation and optimization via CFO under two oracle assumptions, leveraging augmented Lagrangian theory (Sec. 5).
- We demonstrate CFO's ability to trade-off reward maximization and constraint satisfaction in both a visually interpretable setting and a high-dimensional molecular design task (Sec. 6).

## 2. Background and Notation

**Flow Models.** Flow-based generative models constitute a prominent class of approaches for transforming a simple base $p^{\text{base}}$ distribution (e.g., Gaussian) into a complex data distribution $p_{\text{data}}$ (Song et al., 2021a;b; Lipman et al., 2023). Formally, a flow is a time-dependent map $\psi : [0, 1] \times \mathbb{R}^d \to \mathbb{R}^d$, where $\psi_t(x_0)$ denotes the position at time $t$ of a sample that started at $x_0$. The trajectory of $x_t$ is governed by a time-dependent velocity field $u : [0, 1] \times \mathbb{R}^d \to \mathbb{R}^d$ through the ordinary differential equation (ODE):

$$\tfrac{\mathrm{d}}{\mathrm{d}t} \psi_t(x_0) = u_t(\psi_t(x_0)), \quad \psi_0(x_0) = x_0. \quad (1)$$

A *generative* flow model defines a continuous-time Markov process $\{X_t\}_{t \in [0,1]}$, by sampling an initial value $X_0 \sim p^{\text{base}}$ and evolving it according to the flow map, $X_t = \psi_t(X_0)$. The terminal state $X_1 = \psi_1(X_0)$ is required to follow the target distribution, i.e., $X_1 \sim p_{\text{data}}$. Equivalently, the flow induces a family of intermediate marginal densities $p_t$ describing the law of $X_t$ at each time $t \in [0, 1]$. We say that a velocity field $u$ generates the probability path $\{p_t\}_{t \in [0,1]}$ if the random variable $X_t = \psi_t(X_0) \sim p_t$ for all $t < 1$. In practice, choosing $p^{\text{base}} = \mathcal{N}(0, I)$ makes sampling tractable while $u_t$ provides the complexity needed to reach $p_{\text{data}}$.

**Flow Matching.** Flow Matching (Lipman et al., 2023) is a simulation-free algorithm to learn a vector field $u_\theta$, such that the induced marginal densities $p_t^{u_\theta}$ coincide with a prescribed probability path $\{p_t\}_{t \in [0,1]}$ and satisfying $p_0^{u_\theta} = p^{\text{base}}$ and $p_1^{u_\theta} = p_{\text{data}}$. Lipman et al. (2023) demonstrate that the Flow Matching and Conditional Flow Matching objectives share identical gradients, ensuring they converge to the same optimal vector field. In practice, this is achieved by introducing a reference flow and regressing the learned field $u_\theta(x_t, t)$ against the reference velocity:

$$\min_\theta \mathbb{E}_{t, p(x_0, x_1)} \left[ \left\| u_\theta(x_t, t) - \tfrac{\mathrm{d}}{\mathrm{d}t} \psi_t^{\text{ref}}(x) \right\|^2 \right]. \quad (2)$$

With an appropriate choice of the reference flow, specifically one that follows a diffusion trajectory, the Flow Matching

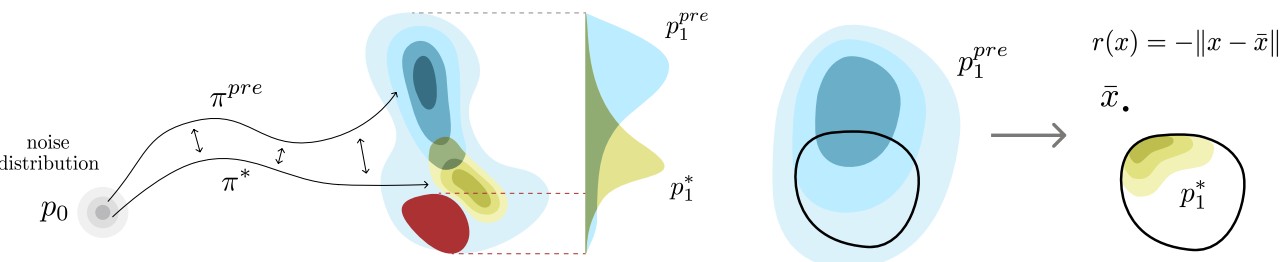

*(a)* Constrained generative optimization via fine-tuning problem.       *(b)* Adaptation to low-cost area within black line.

*Figure 1.* (1a) Pre-trained and fine-tuned policies inducing densities $p_1^{\text{pre}}$ and optimal density $p_1^*$ w.r.t. reward $r$ increasing downwards and in red a high-cost area. (1b) Pre-trained model $p_1^{\text{pre}}$ adapts into $p_1^*$ to maximize $r$ and stay within the constraint region inside the black line.

framework recovers diffusion models as a particular case, showing that diffusion training objectives can be viewed as special instances of flow-based learning (Lipman et al., 2023; Domingo-Enrich et al., 2025). In practice, $u_\theta$ is parameterized by a neural network, and sampling from $p_1^{u_\theta}$ ($\approx p_{\text{data}}$) is performed via simulating the ODE in Eq. 1.

**Reinforcement Learning in continuous-time.** Finite-horizon continuous-time reinforcement learning (RL) (Wang et al., 2020; Treven et al., 2023; Zhao et al., 2025) provides a framework for decision-making in dynamical systems and can be cast as an instance of optimal control. The state space is $\mathcal{X} := \mathbb{R}^d \times [0, 1]$ and actions are taken from an action space $\mathcal{A}$. A policy $\pi : \mathcal{X} \to \mathcal{A}$ prescribes an action for each state $(x, t) \in \mathcal{X}$, yielding the dynamics:

$$\tfrac{\mathrm{d}}{\mathrm{d}t}\psi_t(x) = a_t(\psi_t(x)), \ a_t = \pi(X_t, t), \ X_0 \sim p^{\text{base}}. \quad (3)$$

The resulting process $\{X_t\}_{t\in[0,1]}$ induces a family of marginals $\{p_t^\pi\}_{t\in[0,1]}$. The aim is to optimize the expected performance, typically expressed through an integral reward accumulated along the trajectory and a terminal reward at $t = 1$ (Wang et al., 2020). In our setting, we focus solely on the terminal reward. We use RL notation to emphasize its generality and connection to standard practice, while noting that the setting coincides with deterministic optimal control since both the dynamics and the objective are known.

**Pre-trained Flow Models as RL Policy.** A pre-trained flow can be viewed as a feedback policy: at each time $t$ and state $x$, the velocity field $u^{\text{pre}}(x, t)$ prescribes the action that determines how the system evolves. Defining $a_t = \pi^{\text{pre}}(X_t, t) := u^{\text{pre}}(X_t, t)$ for a policy $\pi^{\text{pre}} : \mathcal{X} \to \mathcal{A}$ (De Santi et al., 2025a), and substituting into Eq. 3, yields deterministic closed-loop dynamics. Starting from $X_0 \sim p_0$, rolling out $\pi^{\text{pre}}$ produces a trajectory $\{X_t\}_{t\in[0,1]}$ with induced marginals $\{p_t^{\pi^{\text{pre}}}\}_{t\in[0,1]}$. Intuitively, the policy selects the direction and speed to steer samples so that their distribution progressively matches the data, with the terminal marginal $p_1^{\text{pre}} := p_1^{\pi^{\text{pre}}} \approx p_{\text{data}}$. Viewing flow models through this policy lens not only unifies flow-based generation and control theory but also enables downstream fine-tuning as policy improvement with a terminal reward. For brevity,

we refer to the pre-trained flow by its implicit policy $\pi^{\text{pre}}$.

## 3. Constrained Generative Optimization via Flow Fine-Tuning

In this work, we aim to fine-tune a pre-trained flow model $\pi^{\text{pre}}$ to obtain a new model $\pi^*$ inducing a process:

$$\tfrac{\mathrm{d}}{\mathrm{d}t}\psi_t(x) = a_t^{\text{fine}}(\psi_t(x)), \ \text{with } a_t^{\text{fine}} = \pi^*(x_t, t). \quad (4)$$

such that its induced distribution $p_1^* := p_1^{\pi^*}$ maximizes the expected value of a property of interest, while satisfying arbitrary constraints and preserving prior information from $\pi^{\text{pre}}$. We denote this problem by *constrained generative optimization via fine-tuning*, illustrated in Figure 1 and defined as:

---

**Constrained Generative Optimization via Flow Fine-Tuning**

$$\arg\max_{\pi} \ \mathbb{E}_{x\sim p_1^\pi}[r(x)] - \alpha D_{KL}(p_1^\pi || p_1^{\text{pre}})$$
$$\text{s.t. } \mathbb{E}_{x\sim p_1^\pi}[c(x)] \leq B \quad (5)$$

---

Where $r : \mathcal{X} \to \mathbb{R}$ and $c : \mathcal{X} \to \mathbb{R}$ are a scalar reward and constraint functions, $\alpha \in \mathbb{R}_+$ determines the KL-regularization strength, and $B \in \mathbb{R}$ is the upper bound on the constraint. At the level of the problem statement, no differentiability of $r$ or $c$ is assumed: Eq. 5 is posed for arbitrary $r, c$. Any additional regularity arises only from the particular inner solver we instantiate later, not from the problem itself. Setting the reward term $r$ to be constant (e.g., $r = 0$) in Eq. 5, reduces the objective to a formulation of *constrained generation* as minimization of a KL divergence between the fine-tuned model density $p_1^\pi$ and the pre-trained model (i.e., $p_1^{\text{pre}}$), while satisfying the expected constraint bound in Eq. 5:

$$\arg\min_{\pi} \ \alpha D_{KL}(p_1^\pi || p_1^{\text{pre}}) \quad \text{s.t.} \quad \mathbb{E}_{x\sim p_1^\pi}[c(x)] \leq B \quad (6)$$

This problem has been studied before in (Chamon et al., 2024; Khalafi et al., 2025). A first approach to tackle Eq. 5 is to optimize a fixed-weight Lagrangian (Chamon et al.,

2024; Zhang et al., 2025):

$$\max_{\pi} \ \mathcal{L}_\mu(\pi) \ = \ \mathbb{E}_{x \sim p_1^\pi}[r(x)] - \alpha D_{KL}(p_1^\pi || p_1^{\text{pre}})$$
$$- \mu \left( \mathbb{E}_{x \sim p_1^\pi}[c(x)] - B \right) \quad \text{s.t. } \mu \geq 0 \tag{7}$$

Here, $\mu \in \mathbb{R}_{\geq 0}$ denotes the Lagrange multiplier that penalizes constraint violations. However, optimizing $\mathcal{L}_\mu$ with a fixed $\mu$ is unreliable for enforcing the constraint. First, feasibility (i.e., $\mathbb{E}_{x \sim p_1^\pi}[c(x)] \leq B$) is not guaranteed for any given $\mu$, unless it exceeds an unknown, problem-dependent threshold. Second, $\mu$ must be tuned by hand, and there is no guaranteed or monotone mapping from $\mu$ to the resulting violation, so trial-and-error often leads to either infeasible or overly conservative solutions. Finally, if $r$ is unbounded or approximate (e.g., a learned proxy reward function), maximizing $\mathcal{L}_\mu$ may shift probability mass toward high-reward regions, yielding invalid designs.

Toward overcoming such limitations, in the next section, we propose an algorithm that can provably tackle the constrained generative optimization problem introduced in Eq. 5 by sequentially fine-tuning the initial pre-trained model via established methods (e.g., Domingo-Enrich et al., 2025).

## 4. Constrained Flow Optimization (CFO)

In the following, we introduce **C**onstrained **F**low **O**ptimization (Alg. 1), which addresses the *constrained generative optimization* problem as formulated in Eq. 5 by solving a sequence of unconstrained entropy-regularized fine-tuning subproblems, each with a different objective function computed via an augmented Lagrangian (AL) scheme (Rockafellar, 1976; Fortin, 1975; Birgin & Martínez, 2014). Intuitively, CFO tackles the problem by embedding the given constraint into an *augmented* reward via two adaptive dual parameters, so that at each iteration, a standard entropy-regularized fine-tuning solver steers the model toward feasibility while improving reward. Concretely, CFO maintains two dual variables, the penalty parameter $\rho_k$ and the Lagrange multiplier $\lambda_k$, whose updates effectively realize a proximal-point-style scheme on the dual variables (Birgin & Martínez, 2014).

**Overview of the Algorithm.** CFO (Alg. 1) takes as input a pre-trained model $\pi_{\text{pre}}$, a number of iterations $K$, a minimal Lagrange multiplier $\lambda_{\min} < 0$, an initial penalty parameter $\rho_{\text{init}} > 0$, a penalty growth rate $\eta \geq 1$, and a contraction value $0 < \tau < 1$. At each iteration $k$, CFO performs 5 main steps:

**Step 1:** An augmented objective $f_k$ (Eq. 8) is formed as the difference between the reward and a penalty term:

$$f_k(x) \ = \ r(x) - \frac{\rho_k}{2} \left[ \max \left( 0, c(x) - B - \frac{\lambda_k}{\rho_k} \right) \right]^2,$$

where the offset $\lambda_k / \rho_k \leq 0$ shifts the term toward the current expected constraint (Birgin & Martínez, 2014).

---

**Algorithm 1** **C**onstrained **F**low **O**ptimization (CFO)

---

1: **Input:** $\pi_{\text{pre}}$: pre-trained model, $K$: number of iterations, $\lambda_{\min} < 0$: min. Lagrange multiplier, $\rho_{\text{init}} > 0$: initial penalty parameter, $\eta \geq 1$: growth rate, $0 < \tau < 1$: contraction value
2: **Init:** Set initial Lagrange multiplier $\lambda_1 = 0$ and penalty $\rho_1 = \rho_{\text{init}}$ parameters
3: **for** $k = 1, 2, \ldots, K$ **do**
4:     **Step 1:** Update fine-tuning AL objective:

$$f_k(x) := r(x) - \frac{\rho_k}{2} \left[ \max \left( 0, c(x) - B - \frac{\lambda_k}{\rho_k} \right) \right]^2 \tag{8}$$

5:     **Step 2:** Compute $\pi_k$ via fine-tuning:

$$\pi_k \leftarrow \text{FineTuningSolver}(f_k, \pi_{\text{pre}}) \tag{9}$$

6:     **Step 3:** Set the empirical constraint gap $G_k$ and contraction statistic $V_k$:

$$G_k = \mathbb{E}_{x \sim p_1^{\pi_k}}[c(x)] - B$$
$$V_k = \min \{ G_k, -\lambda_k/\rho_k \} \tag{10}$$

7:     **Step 4:** Compute Lagrange multiplier proposal:

$$\lambda_{k+1} \leftarrow \max \{ \lambda_{\min}, \min \{ 0, \lambda_k - \rho_k G_k \} \} \tag{11}$$

8:     **Step 5:** Set the new penalty:

$$\rho_{k+1} = \begin{cases} \rho_k, & \text{if } k = 1 \text{ or } V_k \leq \tau V_{k-1}, \\ \eta \rho_k, & \text{otherwise} \end{cases} \tag{12}$$

9: **end for**
10: **Return:** $\pi_K$

---

**Step 2:** A FineTuningSolver (e.g., Domingo-Enrich et al., 2025) computes $\pi_k$ by solving a standard KL-regularized control (or RL) subproblem, with the current objective $f_k$:

$$\pi_k \ \in \ \arg\max_\pi \ \mathbb{E}_{x \sim p_1^\pi}[f_k(x)] - \alpha D_{KL}(p_1^\pi || p_1^{\text{pre}}), \tag{13}$$

For completeness, we report a detailed implementation of this *oracle* step in Appendix A. Other established fine-tuning schemes can be used in place of Adjoint Matching, including gradient-free choices such as DiffusionNFT (Zheng et al., 2025) and Flow-GRPO (Liu et al., 2026), which additionally enable non-differentiable rewards and constraints.

**Step 3:** CFO computes a Monte Carlo estimate of the constraint $c$ under the current policy $\pi_k$ (see Eq. 10), and subtracts the user-defined bound B, thus obtaining the *empirical constraint gap* $G_k$. Furthermore, a *contraction statistic* $V_k$ is computed, which measures the current progress toward feasibility by comparing the recent estimate $G_k$ of the constraint gap with the $\lambda_k / \rho_k \leq 0$ offset term.

**Step 4:** Next, CFO uses the empirical constraint gap $G_k$ to apply a projected dual update to the Lagrange multiplier (Eq. 11). If $G_k > 0$, i.e., the constraint is violated, the multiplier $\lambda_{k+1}$ is decreased, this strengthens the penalty. Instead,

if $G_k < 0$, i.e., the constraint is fulfilled, the Lagrange multiplier $\lambda_k$ is increased toward 0 to relax the penalty strength.

**Step 5:** The contraction statistic $V_k$ (Eq. 10) assesses progress toward feasibility. If $V_k$ does not contract sufficiently, i.e., $V_k > \tau V_{k-1}$, where $\tau$ is a user-defined contraction rate, then CFO infers that the penalty is not sufficiently high and thus increases it by a factor $\eta$. Instead, if $V_k$ is contracting, $\rho$ is kept fixed (Eq. 12). Ultimately, CFO returns the fine-tuned policy $\pi_K$.

A discussion of hyperparameters appears in Appendix D. Nevertheless, it is a priori unclear whether CFO is guaranteed to solve the *constrained generative optimization* problem (Eq. 5). In the next section, we answer this affirmatively by showing that, under oracle assumptions, CFO achieves reward optimality and arbitrary constraint satisfaction.

# 5. Guarantees for CFO

Before presenting the convergence properties of CFO, we first establish a mild and realistic assumption on the FINETUNINGSOLVER used in Alg. 1, which formalizes the approximate nature of its optimization steps and serves as the foundation for the theoretical guarantees that follow.

**Assumption 5.1** (Approx. Solver). *At every iteration $k$, the solver outputs a policy $\pi_k$ satisfying:*

$$L_{\rho_k}(\pi_k, \lambda_k) \geq L_{\rho_k}(\pi, \lambda_k) - \epsilon_k, \quad \forall \pi \qquad (14)$$

*where* $L_{\rho_k}(\pi_k, \lambda_k) = \mathbb{E}_{x \sim p_1^{\pi_k}}[f_k(x)] - \alpha D_{KL}(p_1^\pi || p_1^{\text{pre}})$ *and the sequence* $\{\epsilon_k\} \subseteq \mathbb{R}_+$ *is bounded.*

This assumption captures the approximate nature of practical fine-tuning or optimization oracles, and is standard in augmented Lagrangian (AL) frameworks. It has been adopted in recent works (e.g., De Santi et al., 2025a). The key requirement is that the approximation error remains bounded.

We define the infeasibility of a policy $\pi$ as:

$$G(\pi) = \mathbb{E}_{x \sim p_1^\pi}[c(x)] - B. \qquad (15)$$

If the infeasibility $G(\pi)$ of a given policy is positive, the policy is infeasible, i.e., its average constraint is larger than the permissible bound. If $G(\pi)$ is negative, the policy is feasible and thus fulfills the constraint. Using Assumption 5.1 and Eq. 15, we state our main convergence results for CFO. The proofs are in Appendix E and draw on the analysis developed by Birgin & Martínez (2014).

**Theorem 5.2** (Feasibility of CFO). *Let $\{\pi_k\}$ be a sequence generated by Alg. 1 under Assumption 5.1 on the* FINETUNINGSOLVER. *Let $\bar{\pi}$ be a limit of the sequence $\{\pi_k\}$. Then, we have:*

$$\langle G(\bar{\pi}) \rangle_+ \leq \langle G(\pi) \rangle_+ \quad \forall \pi \qquad (16)$$

*where $G(\pi)$ is defined in Eq. 15 and $\langle \cdot \rangle_+ := \max\{0, \cdot\}$*

Concretely, Theorem 5.2 states that CFO returns a policy that minimizes the introduced infeasibility measure (Eq. 15). Thus, returning either a feasible policy or one which minimizes the infeasibility as far as possible.

**Corollary 5.3** (Feasibility of the Limiting Policy). *Under the same conditions as Theorem 5.2, if the underlying problem admits a feasible policy, then the limiting policy $\bar{\pi}$ is feasible, i.e., it satisfies the constraint (i.e., $G(\bar{\pi}) \leq 0$).*

Theorem 5.2 and Corollary 5.3 establish constraint satisfiability of CFO but do not yet show optimality of the returned policy. To achieve optimality, CFO requires a stronger assumption on the FINETUNINGSOLVER, namely that the approximation error vanishes asymptotically, i.e., $\epsilon_k \to 0$.

**Theorem 5.4** (Optimality of CFO). *Let $\{\pi_k\}$ be the sequence generated by Alg. 1 under Assumption 5.1 with $\lim_{k \to \infty} \epsilon_k = 0$ (see Eq. 14). Let $\bar{\pi}$ be a limit of the sequence $\{\pi_k\}$. If the problem in Eq. 5 is feasible, i.e., $\langle G(\bar{\pi}) \rangle_+ = 0$, then the limiting policy $\bar{\pi}$ is a global maximizer.*

In practice, a FINETUNINGSOLVER achieving $\epsilon_k \to 0$ is rarely available, and the attainable error bounds required by our guarantees depend strongly on the experimental setting. Nevertheless, we show in our experiments (Sec. 6) that using Adjoint Matching (Domingo-Enrich et al., 2025) is sufficient in practice, consistently yielding near-optimal rewards while satisfying the constraints. The convergence guarantees of CFO do not require $r$ or $c$ to be differentiable. Any differentiability assumptions are inherited from the underlying FINETUNINGSOLVER. Consequently, if a gradient-free FINETUNINGSOLVER is used, such as DiffusionNFT (Zheng et al., 2025) or Flow-GRPO (Liu et al., 2026), then CFO can be applied in settings where $r$ and $c$ are available only through function evaluations.

# 6. Experimental Evaluation

We demonstrate the ability of **C**onstrained **F**low **O**ptimization (Alg. 1) to solve the *constrained generative optimization* problem (see Eq. 5) on both low-dimensional illustrative settings, and on molecular design tasks. In particular, we evaluate: (i) the performance of CFO to solve Problem 5 given visually interpretable reward and constraint functions, also for (ii) the sub-case of constrained generation, recovered via a constant reward (see Eq. 6). We further show that (iii) CFO scales to high-dimensional molecular design tasks, and that (iv) it shows promising performances even with an approximate FINETUNINGSOLVER, or when run with a limited number of iterations $K$.

**CFO reliably solves constrained generative optimization low-dimensional tasks.** We evaluate CFO to solve the *constrained generative optimization* problem (Eq. 5) on a visually interpretable setting, where $p_1^{\text{pre}}$ is a mixture of two non-overlapping Gaussians as shown in Figure 2a, enabling

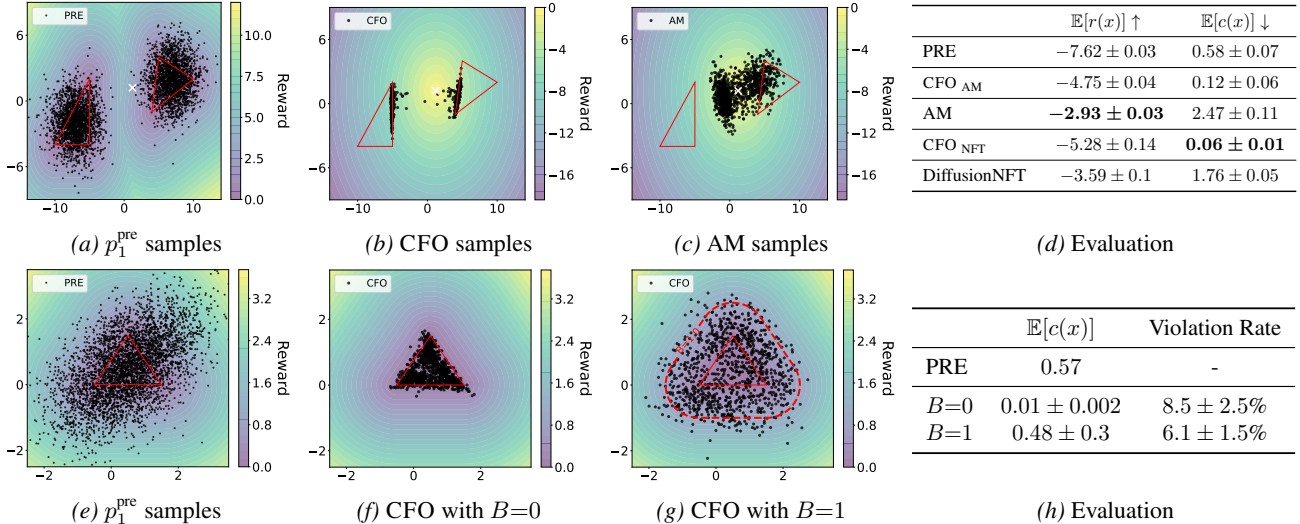

*Figure 2.* **Top** Constrained Generative Optimization: Samples from the pre-trained policy (2a) and policies fine-tuned with CFO (with AM as FINETUNINGSOLVER) (2b) and Adjoint Matching (2c) (AM) (Domingo-Enrich et al., 2025). **Bottom** Constrained generation: Samples from the pre-trained policy (2e) and policies fine-tuned with CFO for $B=0$ (2f) and $B=1$ (2g). The constraint-free area is inside the red triangles. Tables 2d and 2h present numerical results for their row.

visualization of constraint satisfaction during fine-tuning. In this setting, the reward $r$ is the negative squared distance to the white cross in Figures 2a-2c (color-coding in Figure 2b and 2c). The constraint $c$ is zero within the red triangles in Figures 2a-2c, and increases linearly outside (color-coding in Figure 2a). As shown in Figure 2b, CFO, run with $K=20$, and $\rho_{init}=0.5$, steers the pre-trained flow model so its induced density $p^*$ is located predominantly within the valid regions (i.e., red triangles) where the constraint is fulfilled, while simultaneously optimizing the reward by moving samples toward the inner boundaries of both triangles. CFO increases the mean reward from $-7.62$ to $-4.75$ compared to the base model, while it reduces estimated constraint violations from $0.58$ to $0.12$, as reported in Table 2d. The minor residual violations of CFO, in Figure 2b, are likely due to Monte Carlo approximation errors during fine-tuning. In contrast to CFO, Adjoint Matching (Domingo-Enrich et al., 2025), a well-established reward-guided fine-tuning scheme, which does not take into account any constraint, raises the expected reward to $-2.93$, but significantly degrades the model's ability to satisfy the given constraints, increasing constraint violations from $0.58$ to $2.47$ (Figure 2c). The same pattern holds when swapping the inner FINE-TUNINGSOLVER with a gradient-free alternative, such as DiffusionNFT (Zheng et al., 2025). DiffusionNFT (Zheng et al., 2025) alone increases the reward to $-3.59$ but violates the constraint (1.76), whereas CFO $_{NFT}$ attains $-5.28 \pm 0.14$ reward at $0.06 \pm 0.01$ expected constraint (see Table 2d; qualitative samples in Apx. Figure 7), confirming that CFO transfers cleanly across first-order (AM) and zeroth-order (NFT) solvers. We also benchmarked against DiffOpt (Kong et al., 2024), a recent inference-time constrained-generation method that keeps model weights fixed: on this task, Dif-

fOpt (Kong et al., 2024) exhibits a per-sample violation rate of 10–21% versus 8.5% for CFO, while incurring 44–55× higher per-sample sampling cost.

**Constant reward recovers Constrained Generation.** To illustrate the constrained generation (see Eq. 6) capabilities, we consider a correlated Gaussian base density $p_1^{pre}$, visualized in Figure 2e, and a constraint $c$ penalizing samples outside the red central triangle (see Figure 2e). In the following, we vary the bound $B \in \{0.0, 1.0\}$ (see Eq. 5) to obtain diverse flow models inducing fine-tuned distributions $p^*$. As shown in Figures 2f–2g, increasing $B$ causes the resulting densities to visibly expand beyond the zero-constraint region, illustrating the relaxation of constraint enforcement. Quantitatively, the selected degree of permissible violation (i.e., the value of B) is reflected in the mean constraint violations incurred by the respective flow models, obtained by running CFO with $K=20$, and $\rho_{init}=0.5$. As shown in Table 2h, while setting $B=1$ leads to expected constraint value of $0.01$, choosing $B=1.0$ renders CFO less restrictive, inducing a policy $\pi^*$ with a mean constraint of $0.48$. While the base model exhibits $\mathbb{E}_{p_1^{pre}}[c(x)]=0.57$, the violation decreases to $0.48$ under $B=1.0$ and further to $0.01$ under $B=0.0$. These results illustrate how the choice of $B$ controls tolerance to constraint violations, offering a mechanism to adapt CFO to domain-specific requirements.

**CFO scales to high-dimensional molecular design tasks.** To demonstrate the practical relevance of CFO in high-dimensional settings, we apply CFO to a molecular design task, where satisfying constraints is critical. Specifically, we adapt a pre-trained flow model, FlowMol (Dunn & Koes, 2024), on GEOM Drugs (Axelrod & Gomez-Bombarelli, 2022), and maximize the dipole moment (Minkin, 2012)

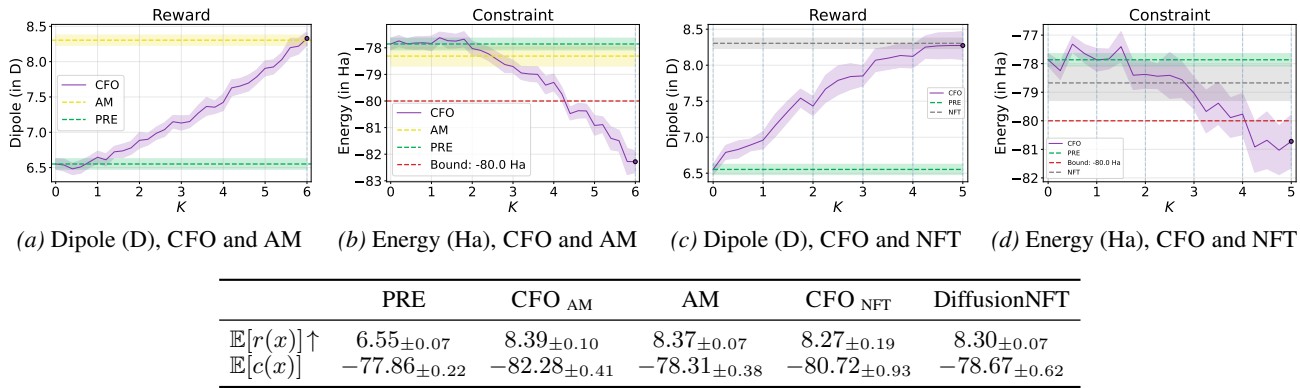

*(a)* Dipole (D), CFO and AM   *(b)* Energy (Ha), CFO and AM   *(c)* Dipole (D), CFO and NFT   *(d)* Energy (Ha), CFO and NFT

|  | PRE | CFO $_{AM}$ | AM | CFO $_{NFT}$ | DiffusionNFT |
|---|---|---|---|---|---|
| $\mathbb{E}[r(x)]\uparrow$ | $6.55_{\pm 0.07}$ | $8.39_{\pm 0.10}$ | $8.37_{\pm 0.07}$ | $8.27_{\pm 0.19}$ | $8.30_{\pm 0.07}$ |
| $\mathbb{E}[c(x)]$ | $-77.86_{\pm 0.22}$ | $-82.28_{\pm 0.41}$ | $-78.31_{\pm 0.38}$ | $-80.72_{\pm 0.93}$ | $-78.67_{\pm 0.62}$ |

*(e)* Evaluation across both FINETUNINGSOLVER choices (95% CI).

*Figure 3.* Energy-constrained dipole moment maximization of FlowMol (Dunn & Koes, 2024) on GEOM Drugs (Axelrod & Gomez-Bombarelli, 2022). CFO attains a dipole moment comparable to the unconstrained baselines, but unlike AM and NFT keeps the expected energy inside the feasible region. (3a-3b): Evolution of the constraint and reward during CFO fine-tuning with ($K=6$, $N=10$) in comparison to AM (Domingo-Enrich et al., 2025) ($N=60$), and we show the final iterate in yellow. (3c-3d): Analogous trajectories with DiffusionNFT (Zheng et al., 2025) as the inner FINETUNINGSOLVER; CFO ($K=5$) drives the expected energy below the $-80$ Ha bound that NFT alone fails to meet, while preserving comparable dipole moment.

under constraint fulfillment. As constraints, we impose an upper bound on the total `xTB` energy (i.e., $-80$ Ha), to be used as a proxy for chemical stability. Further details on the constraint and reward functions employed are provided in Appendix B. Both functions are computed via GNN-based predictors (see Appendix B) trained on `GFN2-xTB` (Bannwarth et al., 2019). We employ differentiable rewards and constraints, because the specific FINETUNINGSOLVER we use in our implementation, namely Adjoint Matching (Domingo-Enrich et al., 2025), requires first-order access to these functions. Our method would also be compatible with non-differentiable rewards and constraints (see Sec. 5).

In Figure 3, we show the performance of CFO for the energy-constrained dipole moment maximization molecular design task. The optimal policy $\pi^*$ computed by CFO ($K=6$, $N=10$) increases the dipole moment from 6.55 Debye of the pre-trained model to 8.39 Debye (Figure 3a). Simultaneously, $\pi^*$ shifts the flow model density to generate predominantly low-energy samples, effectively achieving an expected energy of $-82.28$ Ha, thus satisfying the upper bound B of $-80$ Ha. In Figure 4a-4c, we present drug-like samples from the fine-tuned model, together with their ground-truth reward and constraint values. For reference, running Adjoint Matching ($N=60$) (Domingo-Enrich et al., 2025) purely for reward maximization, without enforcing the constraint, achieves a similar reward of 8.37 Debye, yet results in an expected energy of $-78.31$ Ha, thus not fulfilling the constraint (see Table 3e). In Appendix B, we show that the GNN predictors are accurate throughout the optimization, with ground truth values of reward and constraint being optimized to the same extent.

To showcase that CFO is agnostic to the inner fine-tuning solver, we additionally run CFO with DiffusionNFT (Zheng

et al., 2025) ($K=5$) (Figures 3c-3d). CFO with NFT increases the dipole moment from 6.55 to 8.27 Debye while reducing the expected energy to $-80.72$ Ha, thus satisfying the $-80$ Ha bound. In contrast, unconstrained NFT alone achieves a comparable reward of 8.30 Debye but only $-78.67$ Ha in expected energy, violating the constraint. This confirms that CFO transfers across both first-order (AM) and gradient-free (NFT) solvers, while consistently steering the policy into the feasible region.

Optimizing molecular properties reduces validity, from 34% for PRE to 9% for CFO and to 4% for AM. Since validity is not explicitly enforced but only implicitly learned, fine-tuning steers the model toward sparsely represented regions of chemical space where this notion degrades under our stringent validity criteria (see definition in Apx. C); CFO still maintains higher validity than AM at comparable reward. In Appendix C, we discuss how base model improvements and differentiable geometry relaxation could increase the validity of generated molecules. We also report standard molecular statistics for CFO and AM to contextualize reward-guided fine-tuning. Although not optimization targets, these metrics reflect shifts from the initial model, which is trained on GEOM Drugs and then fine-tuned to maximize the dipole under energy constraints. As shown in Table 4d, CFO exhibits slightly smaller shifts compared to AM, e.g., a QED of 0.37 for CFO, which is closer to the 0.45 of PRE than the 0.34 for AM. Beyond the expectation, we also report per-sample feasibility: CFO satisfies the energy constraint on 61.4% of generated molecules, compared to 40.6% for unconstrained AM. To further illustrate CFO's flexibility, in Appendix C we report an additional experiment where the energy constraint is replaced by a learned `PoseBusters`-based validity criterion (Buttenschoen et al., 2024), which CFO uses to reduce the predicted

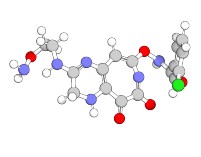

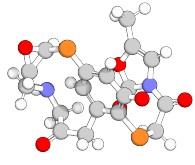

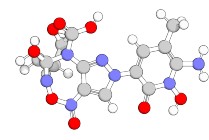

| Feature | PRE | CFO$_{AM}$ | AM | CFO$_{NFT}$ | DiffusionNFT |
|---|---|---|---|---|---|
| Validity (%) | 34 | 9 | 4 | 16 | 17 |
| Sanitization (%) | 47 | 13 | 8 | 21 | 23 |
| QED | 0.45 | 0.37 | 0.34 | 0.28 | 0.29 |
| Lipinski (%) | 88 | 77 | 71 | 49 | 51 |
| logP | 0.89 | $-0.51$ | $-0.95$ | $-3.00$ | $-2.70$ |

*(a)* 15.7 D / $-86.9$ Ha      *(b)* 9.1 D / $-83.4$ Ha      *(c)* 12.5 D / $-93.8$ Ha      *(d)* Molecular Statistic

*Figure 4.* (4a-4c) Drug-like molecules sampled from the fine-tuned model, together with ground-truth dipole moments (D) and energies (Ha). (4d): Molecular statistics for 2000 molecules sampled from polices fine-tuned with CFO and AM. Validity (Definition in Apx. C), RDKit-Sanitization (Landrum, 2025), QED (Ertl & Schuffenhauer, 2009), Lipinski's rule of 5 (Lipinski, 2004), logP (National Center for Biotechnology Information, 2010). CFO preserves FINETUNINGSOLVER-level molecular statistics (e.g., QED, Lipinski) while additionally satisfying the energy budget, i.e., constraint satisfaction comes at essentially no cost in chemical quality compared to pure reward maximization.

violation rate from 53% to 39% while still increasing the dipole moment.

**CFO outperforms a manually tuned penalty baseline.** We compare CFO to a manually tuned fixed-$\mu$ penalty baseline, which optimizes the Lagrangian with a fixed constraint weight $\mu$ (Eq. 7). To select $\mu$, we run the baseline over 18 values covering 13 orders of magnitude, $\mu \in [10^{-6}, 10^{6}]$ (Figure 5a). We observe high sensitivity to $\mu$. For small $\mu$ (e.g., $\mu \leq 0.01$), the baseline attains high reward but exhibits severe constraint violations (e.g., 8.34 D, $-78.94$ Ha for $\mu = 0.01$) and fails to satisfy the constraint. Conversely, for large $\mu$ ($\geq 1.0$), the constraint is enforced, but reward degrades substantially (6.69 Debye for $\mu = 50.0$), falling below CFO. In contrast, CFO satisfies the constraint across all tested hyperparameter settings, while the achieved reward remains stable around 8.39 Debye (ablation study in Apx. D). Overall, only two $\mu$ out of 18 values achieve a reward comparable to CFO while satisfying the constraint (Figure 5c), confirming that manual tuning is unreliable and inefficient (Sec. 3). Since both methods use the same number of gradient steps per run, this exhaustive tuning makes the baseline about $18\times$ more expensive. In contrast, CFO adapts the parameters online, yielding a more robust trade-off between reward maximization and constraint satisfaction.

**CFO can run with approximate fine-tuning oracles and a limited number of iterations $K$.** While CFO performs $K$ outer iterations, standard fine-tuning solvers (Domingo-Enrich et al., 2025; Uehara et al., 2024c) require $N$ inner steps. To avoid a double loop, we fix the total solver budget to $M = K \cdot N$ in all experiments. Increasing $K$ reallocates compute from a more accurate inner solver to more frequent dual updates, trading solver precision for update frequency.

Under a fixed budget $M = 6000$ (Figures 2a–2c), varying $K$ shows a clear reward-constraint trade-off. Few updates ($K = 3$, $N = 2000$) yield high reward but large constraint violation (0.40), while frequent updates ($K = 100$, $N = 60$) nearly eliminate violations (0.10) at the cost of reward ($-5.91$). An intermediate setting ($K = 20$) achieves both

low violation (0.12) and high reward ($-4.75$), see Figure 6. Overall, CFO effectively allocates a fixed compute budget, balancing solver accuracy and dual update frequency, to match the computational cost of the FINETUNINGSOLVER.

Importantly, this observation also holds for the molecular design task in Figure 3. CFO ($K = 6$, $N = 10$) and AM ($N = 60$) have comparable computational cost, as both perform 60 gradient steps. Concretely, CFO has a total runtime of 44.5 min and compares well to the runtime of AM with 40.25 min. This 5% increase arises from the extra sampling and constraint evaluation performed in Step 3 of Alg. 1. Thus demonstrating that CFO can operate effectively in high-dimensional domains even with an approximate oracle. In Appendix C, we additionally report molecular-design results on QM9 (Ramakrishnan et al., 2014) using a differentiable simulator (dxTB (Friede et al., 2024)) as exact reward and constraint functions, complementing the GEOM Drugs experiments shown here.

## 7. Related Work

**Control-based fine-tuning of flow and diffusion models.** Recent works have tackled fine-tuning of diffusion and flow models to maximize expected rewards under KL regularization as an entropy-regularized optimal control problem (e.g., Uehara et al., 2024b; Tang & Zhou, 2024; Uehara et al., 2024c; Domingo-Enrich et al., 2025). Such methods have been successfully applied to real-world domains such as image generation (Domingo-Enrich et al., 2025), molecular design (Uehara et al., 2024c), or protein engineering (Uehara et al., 2024c). These methods have also been adopted as subroutines to tackle settings beyond reward maximization, such as manifold exploration (De Santi et al., 2025a; 2026b) or optimization of distributional objectives, such as conditional value at risk (De Santi et al., 2025b; Wang et al., 2026) or reward-guided model merging (De Santi et al., 2026a). CFO extends fine-tuning methods for reward maximization to leverage known constraint functions and can be straightforwardly used as a plug-in oracle in more

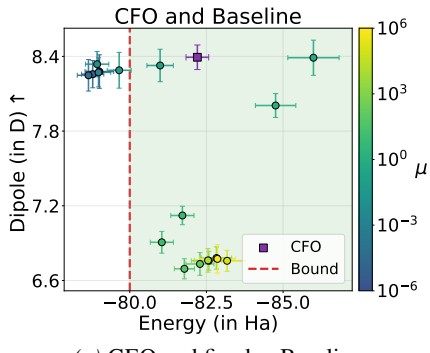

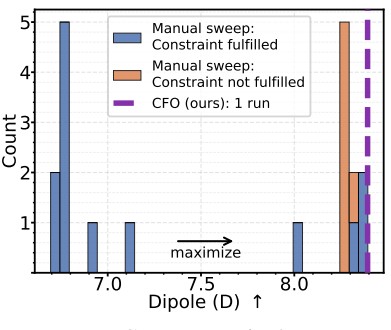

| | $\mathbb{E}[r(x)] \uparrow$ | $\mathbb{E}[c(x)] \leq B$ |
|---|---|---|
| PRE | $6.55_{\pm 0.07}$ | ✗ |
| CFO | $8.39_{\pm 0.10}$ | ✓ |
| $AM_{\mu=0.5}$ | $8.38_{\pm 0.14}$ | ✓ |
| $AM_{\mu=0.1}$ | $8.33_{\pm 0.13}$ | ✓ |
| **Total Runtime** | | |
| CFO | $\approx 44$ min | |
| $AM_{\mu}$ | $\approx 727$ min | |

*(a)* CFO and fixed $\mu$-Baseline     *(b)* Evaluation     *(c)* Counts vs. Dipole

*Figure 5.* (5a-5b): Pareto comparing CFO ($K = 6$, $N = 10$) against fixed-$\mu$ baselines (Eq. 7) run with AM ($N = 60$) (i.e., same number of gradient steps). The baseline with 18 values uniformly across $\mu \in [1e-6, 1e6]$. Manual $\mu$-tuning is unreliable: out of 18 values of $\mu$, only 2 simultaneously achieve CFO-level dipole moments and satisfy the energy constraint. Histogram shows dipole moments for all baseline runs, separated (in color) by feasibility compared to CFO, indicated by the purple dashed line which is feasible without any tuning. 5b: Numeric Evaluation of (5a), (95% CI) $B = -80$ Ha. 5c: Distribution of dipole moments for fixed-$\mu$ baselines separated by constraint satisfaction, CFO result indicated.

complex settings (e.g., exploration and distributional fine-tuning). Importantly, CFO is agnostic to the underlying data modality (continuous, discrete, or mixed): the choice of inner FINETUNINGSOLVER determines this. First-order solvers such as Adjoint Matching (Domingo-Enrich et al., 2025) are well-suited to differentiable rewards and constraints, while gradient-free schemes such as DiffusionNFT (Zheng et al., 2025) and Flow-GRPO (Liu et al., 2026) enable non-differentiable objectives, and discrete-space solvers such as DRAKES (Wang et al., 2025) or SEPO (Zekri & Boullé, 2026) unlock direct application to fully discrete generative models (e.g., for peptide or protein design).

**Constrained Generative Modeling and Optimization.** Most prior work addresses constraint-aware generative modeling, developing tools for handling linear (Graikos et al., 2024), differentiable (Khalafi et al., 2024), and black-box (Kong et al., 2024) constraints. Enforcement spans training-time dual/penalty formulations (Khalafi et al., 2024) and inference-time strategies such as reward-weighted denoising for non-differentiable objectives (Kong et al., 2024) and classifier or classifier-free guidance for differentiable surrogates (Dhariwal & Nichol, 2021; Ho & Salimans, 2021). These techniques have been applied in domains such as molecular design (Kong et al., 2024) and constrained planning (Ma et al., 2025). The closest work to ours is arguably (Khalafi et al., 2024), with the main difference that our setting is for post-training, i.e., at fine-tuning time, constrained generative optimization rather than a training-time scheme enforcing given constraints. Concretely, Khalafi et al. (2024) keep the pre-trained model weights fixed and instead enforce the constraint through inference-time guidance, whereas CFO *fine-tunes* the model so the constraint is internalized into the weights, and inference proceeds as standard sampling at base-model cost.

**Augmented Lagrangian and Dual Methods in Con-**

strained Sampling. Augmented Lagrangian and dual formulations turn equality and inequality constraints into auxiliary updates that run with the sampler, enabling draws from unnormalized targets while enforcing feasibility either per-sample or in expectation (Khalafi et al., 2025; Blanke et al., 2025; Chamon et al., 2024; Smith et al., 2025). For example, Zhang et al. (2025) employ an augmented Lagrangian method to steer diffusion rollouts toward time-varying safety sets without retraining of the base model. Dual schemes similarly maintain physical invariants during sampling or data assimilation while still retaining sufficient exploration of feasible states (Blanke et al., 2025). In addition to constraint generation or sampling, CFO also performs reward-driven optimization with the augmented formulation.

# 8. Conclusion

This work tackles the problem of *constrained generative optimization* via fine-tuning of pre-trained flow and diffusion models, a relevant and challenging task in discovery applications such as drug discovery. After proposing a constrained optimization formulation of the problem, we introduced **C**onstrained **F**low **O**ptimization, a method that transforms the constrained objective into a sequence of fine-tuning steps, and provides feasibility and optimality guarantees. Empirical results on both illustrative settings and molecular design tasks demonstrate the ability of CFO to steer pre-trained flow models toward high-reward regions while satisfying the given constraints. Promising directions include adding zero-order oracles to CFO beyond the current first-order choice, and developing inference-time constraint handling rather than fine-tuning, and testing on protein engineering tasks.

## Acknowledgments

This publication was made possible by the ETH AI Center doctoral fellowship to Riccardo De Santi. The project has received funding from the Swiss National Science Foundation under NCCR Catalysis (grant number 180544 and 225147) and NCCR Automation (grant agreement 51NF40 180545), a National Centre of Competence in Research funded by the Swiss National Science Foundation. This work was supported by an ETH Zurich Research Grant.

## Impact Statement

This paper presents work whose goal is to advance the field of generative optimization. There are many potential societal consequences of our work, none which we feel must be specifically highlighted here.

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

# A. Implementation of FINETUNINGSOLVER- Adjoint Matching (Domingo-Enrich et al., 2025)

To ensure completeness, below we provide pseudocode for one concrete realization of a FINETUNINGSOLVER as in Eq. 9. We describe exactly the version employed in Sec. 6, which builds on the Adjoint Matching framework (Domingo-Enrich et al., 2025), casting linear fine-tuning as a stochastic optimal control problem and tackling it via regression.

Let $u^{\text{pre}}$ be the initial, pre-trained vector field, and $u^{\text{finetuned}}$ its fine-tuned counterpart. We also use $\bar{\alpha}$ to refer to the accumulated noise schedule from Ho et al. (2020), effectively following the flow models notation introduced by Adjoint Matching (Domingo-Enrich et al., 2025, Sec. 5.2). The full procedure is in Alg. 2.

---

**Algorithm 2** FINETUNINGSOLVER- Adjoint Matching (Domingo-Enrich et al., 2025)

---

1: **Input:** $N$ : number of iterations, $u^k$ : current finetuned flow vector field, $u^{\text{pre}}$ : pre-trained flow vector field, $\alpha$ regularization coefficient (Eq. 5), $\nabla f$: objective function gradient, $m$ batch size, $h$ step size
2: **Init:** $u^{\text{finetuned}} := u^k$ with parameter $\theta$
3: **for** $n = 0, 1, 2, \ldots, N - 1$ **do**
4:     Sample $m$ trajectories $\{X_t\}_{0 \leq t \leq 1}$ via a memoryless noise schedule $\sigma(t)$ (Domingo-Enrich et al., 2025), e.g.,

$$\text{sample } \epsilon_t \sim \mathcal{N}(0, I), \ X_0 \sim \mathcal{N}(0, I), \text{ then:} \tag{17}$$

$$X_{t+h} = X_t + h \left( 2u_\theta^{\text{finetuned}}(X_t, t) - \frac{\bar{\alpha}_t}{\alpha_t} X_t \right) + \sqrt{h}\sigma(t)\epsilon_t \tag{18}$$

5:     Use objective function gradient:

$$\tilde{a}_1 = -\frac{1}{\alpha} \nabla_{X_1} f(X_1)$$

6:     For each trajectory, solve the lean adjoint ODE, (Domingo-Enrich et al., 2025, Eq. 38-39), from $t = 1$ to 0:

$$\tilde{a}_{t-h} = \tilde{a}_t + h\tilde{a}_t^\top \nabla_{X_t} \left( 2u^{\text{pre}}(X_t, t) - \frac{\bar{\alpha}_t}{\alpha_t} X_t \right) \tag{19}$$

7:     Where $X_t$ and $\tilde{a}_t$ are computed without gradients, i.e., $X_t = \texttt{stopgrad}(X_t), \tilde{a}_t = \texttt{stopgrad}(\tilde{a}_t)$. For each trajectory, compute the Adjoint Matching objective (Domingo-Enrich et al., 2025, Eq. 37):

$$\mathcal{L}_\theta = \sum_{t \in \{0, h, \ldots, 1-h\}} \left\| \frac{2}{\sigma(t)} \left( u_\theta^{\text{finetuned}}(X_t, t) - u^{\text{pre}}(X_t, t) \right) + \sigma(t)\tilde{a}_t \right\|^2 \tag{20}$$

8:     Compute the gradient $\nabla_\theta \mathcal{L}(\theta)$ and update $\theta$.
9: **end for**
10: **Output:** Fine-tuned flow vector field $u_\theta^{\text{finetuned}}$

---

For further implementation details, we refer to Domingo-Enrich et al. (2025, Appendix G).

# B. Further Experiments and Details - Illustrative Examples

**Reward-only rejection sampling.** We also compare against a simple rejection-sampling baseline, complementary to the fixed-$\mu$ baseline in Eq. 7. We fine-tune a policy purely on the reward signal using Adjoint Matching and then enforce feasibility only by rejecting samples that violate the constraint. On the example in Figure 2c, this reward-only policy attains a constraint satisfaction rate of $13.40\%$, compared to $84.40\%$ for the policy fine-tuned with CFO, i.e., accounting for the constraint during fine-tuning. Inspecting the samples further reveals that (1) violations under CFO occur predominantly near the constraint boundary, and (2) rejection sampling is ineffective when the reward optimum and the constraint region are poorly aligned.

**Details for visually interpretable settings (Figure 2).** The Mixture of Gaussians (Figure 2a) is generated by

$$p(x) = \frac{1}{2}\left(\mathcal{N}\left(x \mid \begin{bmatrix} -7 \\ -2 \end{bmatrix}, \boldsymbol{\Sigma}\right) + \mathcal{N}\left(x \mid \begin{bmatrix} 2 \\ 7 \end{bmatrix}, \boldsymbol{\Sigma}\right)\right), \text{ with } \boldsymbol{\Sigma} = \begin{bmatrix} 3 & 0 \\ 0 & 3 \end{bmatrix},$$

We sample $20k$ points ($80/20$ train/validation split) and train a MLP with 3 hidden layers, each with 256 nodes, for the vector field $v$. The same setting is used for the experiment on the correlated Gaussian (Figure 2e), with:

$$p(x) = \mathcal{N}\left(x \mid \begin{bmatrix} 0.5 \\ 0.5 \end{bmatrix}, \begin{bmatrix} 1 & 0.5 \\ 0.5 & 1 \end{bmatrix}\right)$$

The constraint triangles have the following vertices:

1. **MoG:**

$$\triangle^{I} : \left(\begin{bmatrix} -10 \\ -4 \end{bmatrix}, \begin{bmatrix} -5 \\ -4 \end{bmatrix} \begin{bmatrix} -5 \\ 2 \end{bmatrix}\right) \text{ and } \triangle^{II} : \left(\begin{bmatrix} 4 \\ -1 \end{bmatrix}, \begin{bmatrix} 10 \\ 2 \end{bmatrix}, \begin{bmatrix} 5 \\ 4 \end{bmatrix}\right)$$

2. **Correlated Gaussian:**

$$\triangle : \left(\begin{bmatrix} -1 \\ -0.5 \end{bmatrix}, \begin{bmatrix} 1 \\ -0.5 \end{bmatrix}, \begin{bmatrix} 0 \\ 1 \end{bmatrix}\right)$$

**Computational Cost of: CFO compared to AM.** We plot the computational cost of CFO for a fixed budget $M = 6000$ and a varying $K \in \{3, 15, 20, 100\}$ and compare it to AM (Domingo-Enrich et al., 2025) with $N = 6000$ (see Sec. 6).

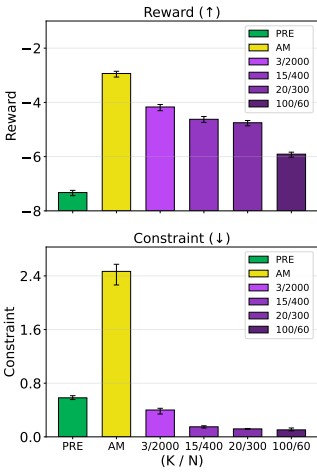

*Figure 6.* Reward and constraint for different values of (K/N)

**Comparison to DiffOpt (Kong et al., 2024), an inference-time constrained-generation method.** DiffOpt is a recent inference-time scheme that keeps the pre-trained model weights fixed and instead modifies the sampling procedure to satisfy constraints (a fundamentally different setting from CFO, which fine-tunes the model). For completeness, we report a comparison on the illustrative MoG task (Figure 2a-2c); the qualitative sample distributions are shown in Figure 7, and numerical results are summarized in Table 1. Across three settings of DiffOpt's guidance and Langevin-MCMC hyperparameters $(\beta_r, \beta_c, L, \eta)$ DiffOpt achieves a per-sample violation rate of $10$–$50\%$ versus $8.5\%$ for CFO, and incurs a

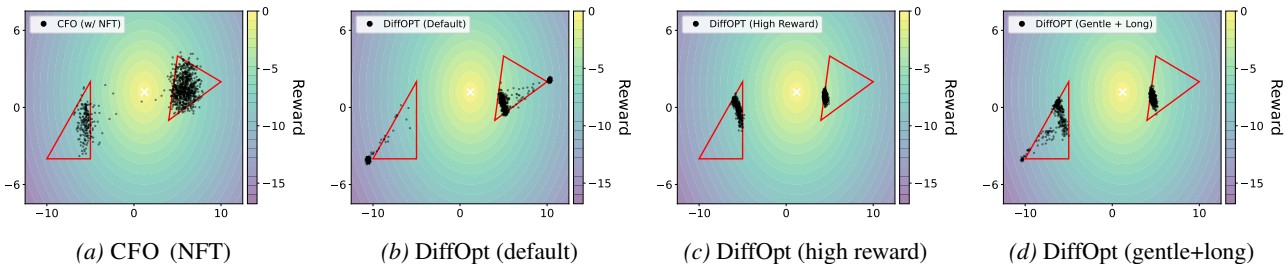

*(a)* CFO (NFT)   *(b)* DiffOpt (default)   *(c)* DiffOpt (high reward)   *(d)* DiffOpt (gentle+long)

*Figure 7.* Qualitative comparison of samples on the MoG task. (7a) CFO with DiffusionNFT (Zheng et al., 2025) as the inner FINETUNINGSOLVER, as a fine-tuning-based reference. (7b-7d) DiffOpt (Kong et al., 2024) under three hyperparameter regimes; see Table 1 for the corresponding numerical results.

*Table 1.* DiffOpt (Kong et al., 2024) on the MoG task across three hyperparameter regimes, compared against CFO. All values are mean $\pm$ 95% CI.

| Method | Settings | Reward $\uparrow$ | Violation $\downarrow$ | KL $\downarrow$ |
|---|---|---|---|---|
| CFO | $K = 20, N = 300$ | $-4.75 \pm 0.04$ | $8.5\%$ | $-$ |
| DiffOpt (default) | $\beta_r = 1, \beta_c = 10, L = 200, \eta = 10^{-3}$ | $-7.54 \pm 0.09$ | $49.4\% \pm 0.6\%$ | $2.05 \pm 0.03$ |
| DiffOpt (high reward) | $\beta_r = 5, \beta_c = 20, L = 500, \eta = 5 \cdot 10^{-4}$ | $-4.52 \pm 0.05$ | $21.4\% \pm 1.7\%$ | $1.48 \pm 0.02$ |
| DiffOpt (gentle+long) | $\beta_r = 2, \beta_c = 15, L = 1000, \eta = 3 \cdot 10^{-4}$ | $-5.05 \pm 0.05$ | $10.4\% \pm 0.6\%$ | $1.31 \pm 0.02$ |

substantially higher per-sample sampling cost (between $44\times$ and $55\times$ slower than CFO) because every sample requires repeated guidance steps. Tuning DiffOpt is therefore essential: *aggressive* settings ($\beta_c \geq 50$) caused sample divergence and were discarded. Even when tuned to either favor reward ("high reward") or constraint satisfaction ("gentle+long"), DiffOpt either violates the constraint substantially (twice the rate of CFO) or, when constraint-tight, sacrifices both reward and sampling efficiency. CFO, on the contrary, fine-tunes the policy once, retains the per-sample inference cost of the base model, and reaches a reward of $-4.75$ at $8.5\%$ violation without any per-sample guidance.

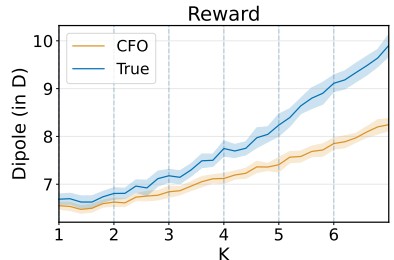
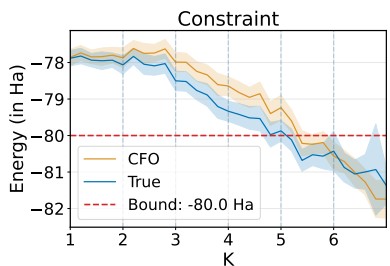

| **MD** | $\mathbb{E}[r(x)] \uparrow$ | $\mathbb{E}[c(x)] \downarrow$ |
|---|---|---|
| True PRE | $6.68 \pm 0.28$ | $-77.71 \pm 0.55$ |
| CFO | $8.37 \pm 0.26$ | $-81.68 \pm 0.71$ |
| xtb | $10.31 \pm 0.46$ | $-81.42 \pm 0.88$ |

*(a)* Dipole Moment (in D)       *(b)* Energy (in Ha)       *(c)* Evaluation

*Figure 8.* Energy-Constrained Dipole Moment Maximization for Molecular Design (MD) (8a-8b): Evolution of the constraint and reward during CFO compared to the true `xtb` Value. 8c: Numeric Comparison between of CFO and `xtb`.

## C. Further Results on Molecular Design Experiments

**Molecular Design.** For the molecular design task, we fine-tune FlowMol (Dunn & Koes, 2024), which jointly models continuous atomic coordinates and discrete categorical variables (atom types, formal charges, bond orders). We refer to Dunn & Koes (2024) for the sampling of categorical and initial values. We use Gaussian sampling for the experiments on GEOM-Drugs and CTMC for the experiments on QM9.

**GNN Details and Generalization.** To verify that optimization targets the intended physical objective rather than exploiting the surrogate, we evaluate the ground-truth `xTB` values for every molecule sampled during the execution of CFO and compare their properties to the GNN predictions. For the energy (used as a constraint), surrogate predictions are essentially indistinguishable from `xTB`, indicating faithful approximation within the explored region. For the dipole moment (the maximization target), the surrogate systematically underestimates the true xTB values by $10\%$, yet the two remain strongly correlated and move in lockstep throughout the fine-tuning. Consequently, improvements under the surrogate translate to larger gains under `xTB`. Overall, these checks indicate that CFO does not exploit model artifacts and remains within the training distribution.

**Additional Results with Exact Rewards and Constraints using `dtxb`.** In a complementary experiment, we employ `dxtb` (Friede et al., 2024) instead of neural approximators to obtain rewards and constraints, which offers exact gradients over atomic positions. For this experiment, we fine-tune FlowMol pre-trained on QM9 (Ramakrishnan et al., 2014). We again maximize the dipole moment while constraining the total energy to remain below $-18$ Ha, a value that differs from the constraint in the main paper due to the different atomic number distribution. As shown in Table 2, the pre-trained model $\pi^{pre}$ violates such a constraint with $65\%$ of samples. In contrast, the model fine-tuned via CFO can successfully achieve zero constraint violation (30 Monte Carlo samples, all below the threshold) while increasing the average norm of the dipole moment from $3.43 \pm 3.45$ to $8.66 \pm 4.50$, as shown in Fig. 9a. As a baseline comparison, we compare to just using Adjoint Matching (Domingo-Enrich et al., 2025), which increases the dipole to 9.04D but also the energy to $-15.5$Ha.

**Results using `posebuster` validity score function.** To further highlight CFO's flexibility, we replace the energy constraint with a molecular-validity criterion based on `posebuster` (Buttenschoen et al., 2024), while keeping the dipole moment as reward. We train a GNN on a custom validation score that equals zero when a molecule is connected and passes the basic `posebuster` checks, and 1 otherwise, running CFO with $K = 2$, $N = 50$, and $B = 0.3$. The pre-trained model attains a dipole moment of 6.92 D but has a $53\%$ constraint-violation rate. In contrast, CFO increases the reward to 9.60 D while reducing the predicted violation rate to $39\%$. In contrast to the energy constraints presented in the main text, the predicted violation rate also differs from the ground truth violation rate, which might be circumvented by an online learning of the constraint function.

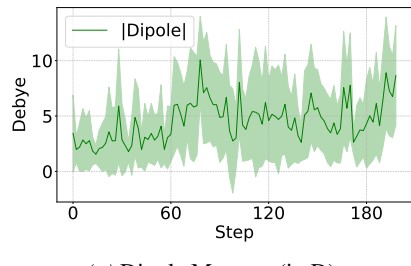

*(a)* Dipole Moment (in D)

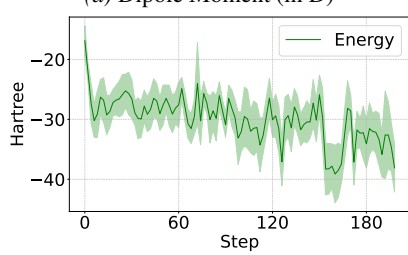

*(b)* Energy (in Ha)

*Figure 9.* Energy-constrained dipole moment maximization on QM9 (Ramakrishnan et al., 2014) and using `dxtb` (Friede et al., 2024) as reward and constraint functions, with exact gradients of the simulation.

**Additional Discussion on Validity of Molecules.** For the molecular design experiments on drug-like molecules presented in the main text, we further apply an RDKit validation step, including stereochemistry reassignment, hydrogen count correction, and full sanitization (valences, kekulization, bond orders). Approximately 7% of final molecules pass, which can be attributed to several reasons: Already in the base FlowMol model, only 34% of molecules fulfill the RDKit validation step, highlighting the need for more diverse pre-training datasets and further base model improvements. Furthermore, the FlowMol-generated geometries used during optimization are not geometrically relaxed, which can lead to invalid bond lengths or angles (see examples in Figure 10). This motivates the development of fully differentiable geometry relaxation methods for molecular design or the extension of CFO to different solvers.

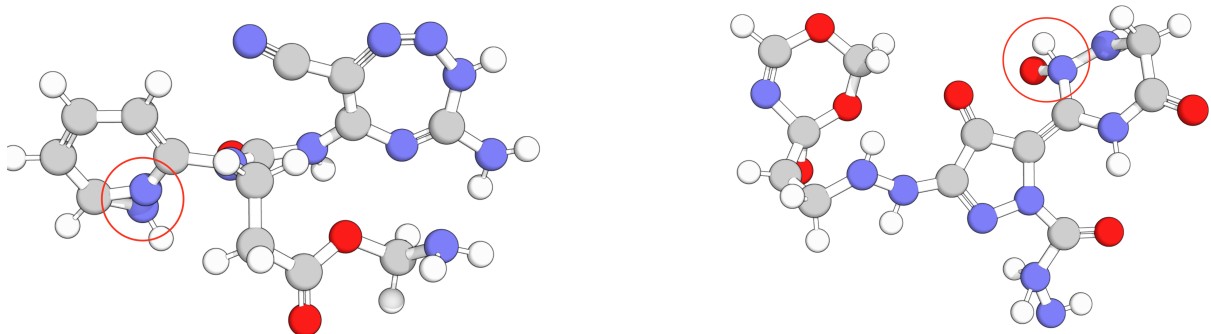

*Figure 10.* Generated drug-like molecules failing the validity test and showing unreasonable bond lengths and angles, highlighted with red circles.

**Definition of stringent validity criteria.** We evaluate validity on a more stringent criterion than pure `RDKit`-Sanitization. Namely, we add two criteria before that to make the workflow as follows:

$$\text{Is sample connected?} \rightarrow \text{Can generated conformer can be embedded?} \rightarrow \text{RDKit} - \text{Sanitization}$$

*Table 2.* Numeric results for CFO on QM9 using `dxtb` for dipole and energy.

| Property | Stage | Value |
|---|---|---|
| Dipole moment | Pre | $3.43 \pm 3.45D$ |
| | CFO | $8.66 \pm 4.50D$ |
| Energy | Pre | $-16.72 \pm 2.48$ Ha |
| | CFO | $-39.40 \pm 4.01$ Ha |
| Violations | Pre | 65% |
| | CFO | 0% |

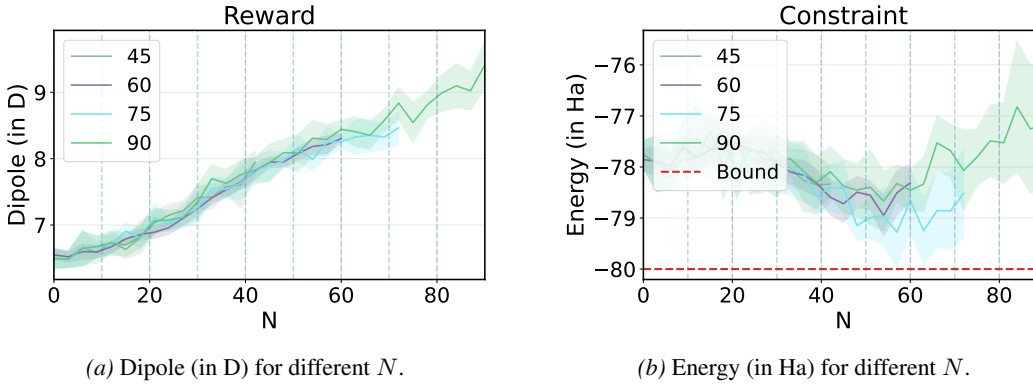

*(a)* Dipole (in D) for different $N$.        *(b)* Energy (in Ha) for different $N$.

*Figure 11.* Unconstrained Dipole maximization of AM (Domingo-Enrich et al., 2025), i.e., $\mu = 0$ in Eq. 7, for different $N$.

## D. Parameter Details, Ablation Studies, and Algorithmic Extensions for CFO and Adjoint Matching

Discussion of the most important Hyperparameters of CFO and FINETUNINGSOLVER:

- **Initial penalty $\rho_{\text{init}}$.** A larger $\rho_{\text{init}}$ penalizes constraint violations more strongly, effectively reducing early exploration within the base distribution. Smaller $\rho_{\text{init}}$ does the opposite.

- **Penalty growth rate $\eta \geq 1$.** Controls the penalty growth across updates. Larger $\eta$ accelerates enforcement and thus can reduce exploration of high-reward regions. Smaller $\eta$ tightens feasibility more gradually, allowing for early reward progress, but potentially slower constraint satisfaction.

- **Contraction rate $\tau \in (0, 1)$.** Determines when to update the penalty parameter $\rho$. Smaller $\tau$ triggers more frequent updates, values near one update conservatively.

- **Multiplier lower bound $\lambda_{\min} < 0$.** Safeguards the Lagrange multiplier via clipping. Smaller $\lambda_{\min}$ permits larger corrective signals of the offset, see Sec. 4. If set to a large negative value, its influence on the final output is typically small, since $\lambda_{\min}$ is not achieved.

- **FINETUNINGSOLVER regularization $\alpha$.** Trade-off between staying close to the base distribution and reallocating mass. Larger $\alpha$ enforces stronger KL-regularization of the policy. A smaller $\alpha$ allows greater deviation from the base policy.

- **Sampling for constraint estimation (sample count/batch size).** Larger samples reduce estimator variance, stabilizing updates and improving feasibility. If the sample size is too small, this yields volatile or biased estimates that can steer CFO to off-target solutions.

**Multi-Constraint Extension of CFO.** CFO straightforwardly extends to multiple constraints $\{c^j\}_{j=1}^J$ with bounds $\{B^j\}_{j=1}^J$ by introducing one Lagrange multiplier $\lambda_k^j$ and penalty $\rho_k^j$ per constraint, yielding the multi-constraint augmented reward

$$f_k(x) = r(x) - \sum_{j=1}^J \frac{\rho_k^j}{2} \left[ \max\left( 0, \ c^j(x) - B^j - \frac{\lambda_k^j}{\rho_k^j} \right) \right]^2.$$

All steps in Alg. 1 apply coordinate-wise to $(\lambda_k^j, \rho_k^j)$, and the feasibility and optimality guarantees of Sec. 5 carry over by standard arguments in the augmented Lagrangian literature (Birgin & Martínez, 2014). This enables, for instance, jointly enforcing an energy constraint and a validity constraint on the molecular design task.

*Table 3.* Hyperparameters for CFO and Adjoint Matching

| | SG(2e-2g) | MoG(2a-2c) | MD-QM9(9a-9b) | MD-GEOM(3a-3b) |
|---|---|---|---|---|
| **CFO** | | | | |
| Lagrangian Updates $K$ | 20 | 20 | 20 | 6 |
| $\rho_{\text{init}}$ | 0.5 | 0.5 | 2 | 1.0 |
| $\eta$ | 1.25 | 1.25 | 1.1 | 1.25 |
| $\tau$ | 0.99 | 0.99 | 0.99 | 0.99 |
| $\lambda_{\text{min}}$ | -50.0 | -50.0 | -50.0 | -50.0 |
| **Adjoint Matching** | | | | |
| $(1/\alpha)$ | 1e5 | 1e5 | 1e2 | 50 |
| Number of Iterations $N$ | 300 | 300 | 10 | 10 |
| Effective Batch Size | 256 | 256 | 40 | 20 |
| Learning Rate | 5e-6 | 5e-6 | 1e-4 | 5e-6 |

**Ablation study for $\rho_{\text{init}}, \eta, \tau$, and $\lambda_{\text{min}}$.** In the following, we provide an ablation study for the molecular design task (Figure 3a-3b) as well as the MoG task (Figure 2b).

*Table 4.* Ablation Study for MoG (2b) and Molecular Design Tasks (3a-3b)

| value | MoG Task | | Molecular Design Task | |
|---|---|---|---|---|
| | $\mathbb{E}[r(x)] \uparrow$ | $\mathbb{E}[c(x)] \downarrow$ | $\mathbb{E}[r(x)] \uparrow$ | $\mathbb{E}[c(x)] \downarrow$ |
| **PRE** | | | | |
| – | $-7.32 \pm 0.08$ | $0.58 \pm 0.02$ | $6.55 \pm 0.07$ | $-77.86 \pm 0.22$ |
| $\rho_{\text{init}}$ | | | | |
| 0.1 | $-4.49 \pm 0.06$ | $0.24 \pm 0.02$ | $8.43 \pm 0.12$ | $-82.21 \pm 0.33$ |
| 1.0 | $-4.88 \pm 0.07$ | $0.10 \pm 0.01$ | $8.36 \pm 0.11$ | $-81.99 \pm 0.45$ |
| 10.0 | $-5.62 \pm 0.09$ | $0.10 \pm 0.01$ | $8.22 \pm 0.08$ | $-81.91 \pm 0.37$ |
| $\eta$ | | | | |
| 1.0 | $-4.56 \pm 0.05$ | $0.17 \pm 0.01$ | $8.33 \pm 0.11$ | $-82.22 \pm 0.39$ |
| 1.25 | $-4.75 \pm 0.06$ | $0.12 \pm 0.01$ | $8.30 \pm 0.12$ | $-81.99 \pm 0.34$ |
| 2.0 | $-5.34 \pm 0.15$ | $0.10 \pm 0.01$ | $8.39 \pm 0.27$ | $-81.98 \pm 0.46$ |
| $\lambda_{\text{min}}$ | | | | |
| 0.0 | $-4.84 \pm 0.04$ | $0.16 \pm 0.01$ | $8.39 \pm 0.10$ | $-81.85 \pm 0.31$ |
| -1.0 | $-4.40 \pm 0.76$ | $0.26 \pm 0.02$ | $8.30 \pm 0.12$ | $-81.80 \pm 0.42$ |
| -10.0 | $-4.75 \pm 0.06$ | $0.12 \pm 0.01$ | $8.26 \pm 0.11$ | $-82.13 \pm 0.39$ |
| -50.0 | $-4.75 \pm 0.06$ | $0.12 \pm 0.01$ | $8.35 \pm 0.13$ | $-82.16 \pm 0.48$ |
| $\tau$ | | | | |
| 0.5 | $-5.02 \pm 0.05$ | $0.10 \pm 0.01$ | $8.34 \pm 0.11$ | $-82.06 \pm 0.32$ |
| 0.75 | $-4.98 \pm 0.05$ | $0.10 \pm 0.01$ | $8.31 \pm 0.13$ | $-82.05 \pm 0.40$ |
| 0.9 | $-4.82 \pm 0.07$ | $0.11 \pm 0.01$ | $8.31 \pm 0.12$ | $-81.93 \pm 0.40$ |
| 0.99 | $-4.75 \pm 0.06$ | $0.12 \pm 0.01$ | $8.39 \pm 0.13$ | $-82.27 \pm 0.33$ |

Across tasks, CFO's sensitivity to hyperparameters varies: while the MoG task exhibits clear shifts in reward and constraint satisfaction across settings, the molecular design task remains highly robust, with only minor fluctuations. Larger initial $\rho_{\text{init}}$ and higher $\eta$ consistently tighten constraint satisfaction at the cost of modestly reduced reward, whereas $\lambda_{\text{min}}$ and $\tau$ have a lower effect. The lower effect of $\lambda_{\text{min}}$ likely stems from $\lambda$ rarely reaching its lower bound, and the contraction parameter barely impacts updates. A separate batch-size ablation on MoG shows that larger batches significantly improve constraint

satisfaction and reward maximization.

*Table 5.* Ablation Study for the MoG task with different batch sizes

| value | $\mathbb{E}[r(x)] \uparrow$ | $\mathbb{E}[c(x)] \downarrow$ |
|---|---|---|
| Batch Size | | |
| 8 | $-5.16 \pm 0.11$ | $0.36 \pm 0.04$ |
| 32 | $-4.93 \pm 0.08$ | $0.27 \pm 0.05$ |
| 128 | $-4.74 \pm 0.06$ | $0.14 \pm 0.02$ |
| 512 | $-4.68 \pm 0.05$ | $0.11 \pm 0.01$ |

# E. Proofs

Before we present a proof of the theorems in Section 5. We will transform the main problem in Eq. 5 to a simpler form. First, we recall that the policy $\pi$ is a vector field. It has been shown before that the ODE in Eq. 1 and a stochastic differential equation (SDE) of the form

$$dX_t = b(X_t, t)dt + \sigma(t)dB_t, \; X_0 \sim p_0, \tag{21}$$

with drift $b : \mathbb{R}^d \times [0,1] \to \mathbb{R}^d$, diffusion coefficient $\sigma : [0,1] \to \mathbb{R}_{\geq 0}$ and Brownian motion $B_t$ induce the same marginals $\{p_t\}$. For an exact definition of $b$ and a proof of this statement, we refer to (Domingo-Enrich et al., 2025). Controlling this SDE can be done by adjusting the drift as follows (Tang & Zhou, 2024; Domingo-Enrich et al., 2025):

$$dX_t = (b(X_t, t) + \sigma(t)u(X_t, t)) \, dt + \sigma(t)dB_t, \; X_0 \sim p_0,$$

where $u : \mathbb{R}^d \times [0,1] \to \mathbb{R}^d$ is a control vector field, this means the pre-trained model is a controlled model with $u \equiv 0$. With these notational changes, we reformulate the optimization problem in Eq. 5 in terms of the controlled diffusion process $\mathbf{X}^u \sim p^u$:

$$
\begin{aligned}
\max_{u \in \mathcal{U}} \quad & \mathbb{E}_{\mathbf{X}^u \sim p^u}\left[r(X_1)\right] - \alpha D_{KL}(p_1^u(\cdot) || p_1^{\text{pre}}(\cdot)) \\
\text{s.t.} \quad & \mathbb{E}_{\mathbf{X}^u \sim p^u}\left[c(X_1)\right] \leq B
\end{aligned}
\tag{22}
$$

Eq. 22 may seem the same as Eq. 5, but it is in terms of a diffusion process. This way we can calculate the KL efficiently, see (Eq. 18, Domingo-Enrich et al., 2025), by using Girsanov's theorem, which gives the relationship between the control process $u$ and the KL-Divergence:

$$D_{\text{KL}}(p^u(\mathbf{X}|X_0) \, || \, p^{\text{pre}}(\mathbf{X}|X_0)) = \mathbb{E}_{\mathbf{X}^u \sim p^u}\left[\int_0^1 \frac{1}{2}\|u(X_t, t)\|^2 \, dB_t\right]$$

Meaning if both processes have the same initial value $X_0$, the KL divergence between the controlled and uncontrolled process is equal to the expected value of the squared norm of the control $u$ (Domingo-Enrich et al., 2025; Uehara et al., 2024b; Tang & Zhou, 2024). This dependence on the initial value can be dropped when using a specific noise schedule (Domingo-Enrich et al., 2025). Recalling that marginals at time $t$ are $p_t(x)$, i.e. $X_t \sim p_t(x)$, then we can equivalently write the optimization problem as:

$$
\begin{aligned}
\max_{u \in \mathcal{U}} \quad & \mathbb{E}_{\mathbf{X}^u \sim p^u}\left[r(X_1)\right] - \alpha \mathbb{E}\left[\int_0^1 \frac{1}{2}\|u(X_t^u, t)\|^2 dt\right] \\
\text{s.t.} \quad & \mathbb{E}_{\mathbf{X}^u \sim p^u}\left[c(X_1)\right] \leq B
\end{aligned}
$$

Where the expectation is taken over the controlled process $\mathbf{X}^u$. For numerical optimization, we now assume that the control $u$ is a parametric model, typically a neural network, with parameters $\theta$. The resulting optimization problem is then:

$$
\begin{aligned}
\max_{\theta \in \mathbb{R}^m} \quad & F(\theta) := F_r(\theta) - \alpha F_{KL}(\theta) \\
& = \mathbb{E}_{x \sim p_1^{u_\theta}}[r(x)] - \alpha \mathbb{E}\left[\int_0^1 \frac{1}{2}\|u_\theta(X_t, t)\|^2 dt\right] \\
\text{s.t.} \quad & G(\theta) := \mathbb{E}_{x \sim p_1^{u_\theta}}[c(x)] - B \leq 0
\end{aligned}
\tag{23}
$$

For some function $F : \mathbb{R}^m \to \mathbb{R}$ and function $G : \mathbb{R}^m \to \mathbb{R}$. This is finite-dimensional optimization over $\theta$.

Next, we present a proof that Alg. 1 can find a parameterized policy $\pi_\theta$, with $\theta \in \mathbb{R}^m$ that minimizes the infeasibility while maximizing the reward. The proof is adapted from "Practical Augmented Lagrangian Methods for Constrained Optimization" (Birgin & Martínez, 2014, Chapter 5).

The augmented Lagrangian objective in Eq. 13 becomes:

$$L_\rho(\theta, \lambda) = F(\theta) - \frac{\rho}{2}\left[\max\left(0, G(\theta) - \frac{\lambda}{\rho}\right)\right]^2 \tag{24}$$

where $\lambda \in \mathbb{R}_{\leq 0}$ is the Lagrange multiplier, $\rho > 0$ is a penalty parameter.

With this notation, the assumption on the solver becomes:

**Assumption E.1** (Solver). For all $k \in \mathbb{N}$, we obtain $u$ such that:

$$L_{\rho_k}(\theta_k, \lambda_k) \geq L_{\rho_k}(\theta, \lambda_k) - \epsilon_k \quad \forall\, \theta \in \mathbb{R}^m \tag{25}$$

where the sequence $\{\epsilon_k\} \subseteq \mathbb{R}_+$ is bounded.

This corresponds to Assumption 5.1 from (Birgin & Martínez, 2014). Assumption E.1 states that the solver can find an approximate maximizer of the subproblem.

Next we state and prove the main result for the algorithm. Namely, in the limit, we obtain a minimizer of the infeasibility measure.

**Theorem E.2** (Feasibility of Constrained Flow Optimization). *Let $\{\theta_k\}$ be a sequence generated by Alg. 1 under the solver Assumption E.1. Let $\bar{\theta}$ be a limit of the sequence $\{\theta_k\}$. Then, we have:*

$$\langle G(\bar{\theta}) \rangle_+ \leq \langle G(\theta) \rangle_+ \quad \forall \theta \in \mathbb{R}^m, \tag{26}$$

*where $G(\theta) := \mathbb{E}_{x \sim p_1^{u_\theta}}[c(x)] - B \leq 0$ and $\langle \cdot \rangle_+ := \max\{0, \cdot\}$.*

*Proof.* By definition $\mathbb{R}^m$ is closed and $\theta_k \in \mathbb{R}^m$ thus $\bar{\theta} \in \mathbb{R}^m$. We consider two cases: $\{\rho_k\}$ bounded and $\rho_k \to \infty$. First we assume $\{\rho_k\}$ is bounded, there exists $k_0$ such that $\rho_k = \rho_{k_0}$ for all $k \geq k_0$. Therefore, for all $k \geq k_0$, the upper bracket of Eq. 12 holds. This implies that $|V_k| \to 0$, so $\langle G(\theta_k) \rangle_+ \to 0$. Thus, the limit point is feasible.

Now, assume that $\rho_k \to \infty$. Let $K \subseteq \mathbb{N}$ be such that:

$$\theta_k \to \bar{\theta} \ \text{ for } \ k \in K \text{ and } k \to \infty$$

Assume by contradiction that there exists $\theta \in \mathbb{R}^d$ such that

$$\langle G(\bar{\theta}) \rangle_+^2 > \langle G(\theta) \rangle_+^2$$

By the continuity of $G$, the boundedness of $\{\lambda_k\}$, and the fact that $\rho_k \to \infty$, there exists $c > 0$ and $k_0 \in \mathbb{N}$ such that for all $k \in K, k \geq k_0$:

$$\left\langle G(\theta_k) - \frac{\lambda_k}{\rho_k} \right\rangle_+^2 > \left\langle G(\theta) - \frac{\lambda_k}{\rho_k} \right\rangle_+^2 + c$$

Therefore, for all $k \in K, k \geq k_0$:

$$F(\theta_k) - \frac{\rho_k}{2}\left[\left\langle G(\theta_k) - \frac{\lambda_k}{\rho_k} \right\rangle_+^2\right] < F(\theta) - \frac{\rho_k}{2}\left[\left\langle G(\theta) - \frac{\lambda_k}{\rho_k} \right\rangle_+^2\right] - \frac{\rho_k c}{2} + F(\theta_k) - F(\theta)$$

Since $\lim_{k \in K} \theta_k = \bar{\theta}$, the continuity of $F$, and the boundedness of $\{\epsilon_k\}$, there exists $k_1 \geq k_0$ such that, for $k \in K\ k \geq k_1$:

$$\frac{\rho_k c}{2} - F(\theta_k) + F(\theta) > \epsilon_k$$

Therefore,

$$F(\theta_k) - \frac{\rho_k}{2}\left[\left\langle G(\theta_k) - \frac{\lambda_k}{\rho_k} \right\rangle_+^2\right] < F(\theta) - \frac{\rho_k}{2}\left[\left\langle G(\theta) - \frac{\lambda_k}{\rho_k} \right\rangle_+^2\right] - \epsilon_k$$

for $k \in K, k \geq k_1$. This contradicts Assumption E.1. $\square$

Theorem E.2 and its proof correspond to (Birgin & Martínez, 2014, Sec. 5.1). Theorem E.2 establishes that Alg. 1, under the iterates given in Assumption E.1, identifies minimizers of the infeasibility, i.e.,

$$\langle G(\theta) \rangle_+ := \left\langle \mathbb{E}_{x \sim p_1^{u_\theta}}[c(x)] - B \leq 0 \right\rangle_+.$$

Consequently, if the original optimization problem is feasible, then every limit point of the sequence produced by the algorithm is also feasible.

Next, we will see that, assuming that $\epsilon_k$ tends to zero, it is possible to prove that, in the feasible case, the algorithm asymptotically finds a global maximizer of the problem in Eq. 5.

**Theorem E.3** (Optimality of Constrained Flow Optimization). *Let $\{\theta_k\} \subset \mathbb{R}^d$ be a sequence generated by Alg. 1 under Assumption E.1 and $\lim_{k\to\infty} \epsilon_k = 0$. Let $\bar{\theta} \in \mathbb{R}^m$ be a limit of the sequence $\{\theta_k\}$. Suppose that $\langle G(\theta) \rangle_+ = 0$, then $\bar{\theta}$ is a global maximizer of Eq. 5.*

*Proof.* Let $K \subseteq \mathbb{N}$ be such that.

$$\theta_k \to \bar{\theta} \text{ for } k \in K \text{ and } k \to \infty$$

By assumption, the problem is feasible, thus, by Theorem E.2, we have that $\bar{\theta}$ is feasible. Let $\theta \in \mathbb{R}^m$ be such that $G(\theta) \leq 0$. By the definition of the algorithm, we have that

$$F(\theta_k) - \frac{\rho_k}{2}\left[\left\langle G(\theta_k) - \frac{\lambda_k}{\rho_k}\right\rangle_+^2\right] \geq F(\theta) - \frac{\rho_k}{2}\left[\left\langle G(\theta) - \frac{\lambda_k}{\rho_k}\right\rangle_+^2\right] - \epsilon_k \tag{27}$$

for all $k \in \mathbb{N}$, as well as by assumption $G(\theta) \leq 0$, we have that

$$\left\langle G(\theta) - \frac{\lambda_k}{\rho_k}\right\rangle_+^2 \leq \left(\frac{\lambda_k}{\rho_k}\right)^2. \tag{28}$$

We again consider the two cases: $\rho_k \to \infty$ and $\{\rho_k\}$ bounded.

In the first case, we assume $\rho_k \to \infty$. By Eq. 27 and Eq. 28, we have

$$F(\theta_k) \geq F(\theta_k) - \frac{\rho_k}{2}\left[\left\langle G(\theta_k) - \frac{\lambda_k}{\rho_k}\right\rangle_+^2\right] \geq F(\theta) - \frac{(\lambda_k)^2}{2\rho_k} - \epsilon_k.$$

Taking limits for $k \in K$, and using that $\theta_k \to \bar{\theta}$, we have that $\lim_{k\in K}(\lambda_k)^2/\rho_k = 0$ and $\lim_{k\in K}\epsilon_k = 0$, by the continuity of $F$ and the convergence of $\theta_k$, we get

$$F(\bar{\theta}) \geq F(\theta).$$

Since $\theta$ is an arbitrary feasible element of $\mathbb{R}^m$, $\bar{\theta}$ is a global optimizer.

For the second case, we assume $\{\rho_k\}$ is bounded, there exists $k_0 \in \mathbb{N}$ such that $\rho_k = \rho_{k_0}$ for all $k \geq k_0$. Therefore, by Assumption E.1, Eq. 27 holds for all $k \geq k_0$, and Eq. 28 holds with $\rho = \rho_{k_0}$. Thus,

$$F(\theta_k) - \frac{\rho_{k_0}}{2}\left[\left\langle G(\theta_k) - \frac{\lambda_k}{\rho_{k_0}}\right\rangle_+^2\right] \geq F(\theta) - \frac{(\lambda_k)^2}{2\rho_{k_0}} - \epsilon_k.$$

for all $k \geq k_0$. Let $K_1 \subseteq \mathbb{N}$ and $\lambda^* \in \mathbb{R}_{\leq 0}$ be such that: $\lim_{k\in K_1} \lambda_k = \lambda^*$. By the feasibility of $\bar{\theta}$, taking limits in the inequality above for $k \in K_1$, we get

$$F(\bar{\theta}) - \frac{\rho_{k_0}}{2}\left[\left\langle G(\bar{\theta}) - \frac{\lambda^*}{\rho_{k_0}}\right\rangle_+^2\right] \geq F(\theta) - \frac{(\lambda^*)^2}{2\rho_{k_0}} - \epsilon_k. \tag{29}$$

Now, if $G(\bar{\theta}) = 0$, since $\lambda^*/\rho_{k_0} \geq 0$, we have that

$$\left\langle G(\bar{\theta}) - \frac{\lambda^*}{\rho_{k_0}}\right\rangle_+^2 = \left(\frac{\lambda^*}{\rho_{k_0}}\right)^2$$

Therefore, by Eq. 29,

$$F(\bar{\theta}) - \frac{\rho_{k_0}}{2}\left[\left\langle G(\bar{\theta}) - \frac{\lambda^*}{\rho_{k_0}}\right\rangle_+^2\right] \geq F(\theta) - \frac{(\lambda^*)^2}{2\rho_{k_0}}. \tag{30}$$

But, by Eq. 10, $\lim_{k\to\infty} \min\{G(\theta_k), -\lambda^*/\rho_{k_0}\} = 0$. Therefore, if $G(\bar{\theta}) < 0$, we necessarily have that $\lambda^* = 0$. Therefore, Eq. 30 implies that $F(\bar{\theta}) \geq F(\theta)$. Since $\theta$ is an arbitrary feasible element of $\mathbb{R}^m$, $\bar{\theta}$ is a global optimizer. $\square$

We want to make two remarks about Theorem E.3: first, as mentioned in Sec. 5, having access to such a solver is difficult and, in practice, rarely the case. Secondly, we refer the reader to (Birgin & Martínez, 2014, Sec. 5.2) for a discussion about the sets $K$ and $K_1$, how they are connected to the convexity of $F$ and $G$, and the corresponding theorem and proof.

