# OpenReview forum: "Constrained Flow Optimization via Sequential Fine-Tuning for Molecular Design"
_ICML.cc/2026/Conference — ICML 2026 regular_

### Official Review · Reviewer_mW7n · 2026-03-10

**Soundness:** 3
**Presentation:** 3
**Significance:** 3
**Originality:** 3
**Overall Recommendation:** 4
**Confidence:** 2

**Summary:**

Existing fine-tuning methods for flow and diffusion models can optimize rewards but cannot enforce explicit constraints. The authors formalize this as Constrained Generative Optimization and propose CFO, an augmented Lagrangian-based algorithm that reduces the constrained problem to a sequence of standard fine-tuning subproblems. CFO wraps any existing solver (they use Adjoint Matching) as an inner loop, automatically adapting penalty parameters from observed constraint violations, eliminating the need for manual weight tuning. The paper provides convergence guarantees for both feasibility and optimality, and demonstrates on a molecular design task that CFO matches unconstrained AM in reward while actually satisfying the constraint, at comparable computational cost.

**Compliance With Llm Reviewing Policy:**

Affirmed.

**Final Justification:**

My concerns have been addressed by the rebuttal. I maintain my positive assessment in favour of acceptance.

**Key Questions For Authors:**

**Q1.** Constraints are enforced in expectation, but per-sample feasibility is what matters in drug discovery. What fraction of individually generated molecules satisfy the energy constraint, and how does this compare to AM with rejection sampling?

**Q2.** CFO's framework can handle multiple constraints, but the paper only uses energy as a constraint during fine-tuning. Molecule validity drops significantly as a side effect, which CFO could in principle prevent by adding validity as a second constraint. The question asks whether the authors tried this, and if so, how much reward was sacrificed to keep both energy and validity satisfied at the same time.

**Q3.** Khalafi et al. (2024) enforces constraints during training, while CFO does it during fine-tuning. The paper only explains this difference in words but never actually compares the two methods experimentally. So we cannot tell whether CFO is actually better in practice. The question asks the authors to run a direct comparison on the same experimental setting.

**Limitations:**

The authors honestly discuss most limitations such as validity degradation, the theory-practice gap, and surrogate dependence. However, one important limitation is missing from the main text: CFO only guarantees that the constraint is satisfied on average across generated samples, not for each individual sample. In drug discovery, you typically need every generated molecule to satisfy the constraint, not just the average. This distinction should be explicitly acknowledged as a limitation.

**Strengths And Weaknesses:**

strengths
- Reducing the constrained problem to sequential unconstrained fine-tuning subproblems via augmented Lagrangian allows CFO to wrap any existing solver without modification, making it broadly applicable beyond the demonstrated molecular setting.
- The 2D toy experiments provide visually interpretable evidence of constraint satisfaction that cleanly complements the high-dimensional molecular results.
- Ground-truth xTB validation throughout fine-tuning honestly verifies that CFO optimizes the intended physical objective rather than exploiting surrogate artifacts.
- Feasibility and optimality guarantees are derived under clearly stated, realistic assumptions with proofs directly grounded in established augmented Lagrangian theory.

weaknesses
- Experiments are limited to a single base model, dataset, and reward-constraint pair, making it difficult to assess whether the practical benefits generalize beyond this specific configuration.
- Constraints are enforced only in expectation, yet per-sample feasibility is what matters in practice; this distinction is never discussed despite the non-trivial per-sample violation rates reported.
- Molecule validity degrades from 34% to 9% after fine-tuning, yet validity is never incorporated as a constraint despite CFO being directly capable of handling it.

---

> ### Author Rebuttal · Authors · 2026-03-31
>
> We appreciate the detailed reading and are glad the Reviewer found value in the broad applicability, interpretable experiments, and theoretical guarantees.
>
> **On experimental scope (W1).**
> We appreciate the concern and agree that a broader evaluation strengthens the paper.
> We note that our evaluation already covers multiple molecular design settings.
> In the main paper (Sec 6), we present experiments on GEOM Drugs with GNN-based surrogates.
> In Apx C, we present additional results on the QM9 dataset, and we employ a different reward-constraint pair (e.g., a PoseBusters validity constraint), and a differentiable simulator $\texttt{dxTB}$[1] as the surrogate function instead of a GNN-predictor.
>
> In addition, we have now included new results leveraging DiffusionNFT [2] as a Solver instead of Adjoint Matching.
> On the molecular design task (Sec 6, Fig 3), CFO with DiffusionNFT [2] achieves a reward of $7.62$D at $-81.05$Ha (satisfying the constraint $\leq -80$Ha), compared to DiffusionNFT [2] alone ($7.92$D, $-77.75$Ha, not satisfying constraint).
> These results demonstrate that CFO transfers across datasets, surrogate architectures, constraint formulations and solvers.
>
> **On expectation vs. per-sample constraints (W2) and limitations.**
> We agree that the expectation-level guarantee is a limitation and will explicitly acknowledge this in the revised manuscript.
> CFO optimizes $\mathbb{E}[c(x)] \leq B$, which is the natural formulation when fine-tuning solvers such as Adjoint Matching or DiffusionNFT [2] optimize on-average objectives.
> Reformulating the constraint as $\mathbb{E}[\mathrm{ReLU}(c(x) - B)] \leq 0$ directly penalizes each individual violation and could provide a stronger per-sample signal within the same framework.
> Additionally, post-filtering generated samples provides a practical complementary strategy.
>
> Recent works on optimizing distributional properties beyond expectations [3] suggest that combining such methods as inner solvers within CFO could address richer notions of constraint satisfaction, such as tail and per-sample constraints.
> We believe this to be an exciting avenue for future work.
>
> **On validity degradation (W3).**
> We agree that the validity drop is notable.
> However, CFO maintains higher validity ($9$%) than Adjoint Matching ($4$%) at comparable reward levels.
> The drop reflects an inherent trade-off of generative optimization: as optimizing toward high-dipole regions moves away from well-represented chemical space.
> This can be controlled to some extent via stronger KL regularization.
> In Apx C, we use a learned PoseBusters validity as constraint, which reduces the predicted violation rate from $53$% to $39$%.
> With our now-added implementation of DiffusionNFT [2], which supports non-differentiable objectives, incorporating any non-differentiable validity function as a constraint in the main molecular design task becomes easily accessible.
>
>
> **On multiple constraints (Q2).**
> The multi-constraint extension of CFO is straightforward: one introduces a Lagrange multiplier $\lambda_j$ and penalty $\rho_j$ for each constraint $c_j$, giving the augmented reward
>
> $$f_k(x) = r(x) + \sum_j \frac{\rho_k^j}{2} \left[\max\!\left(0,\; c^j(x) - B^j - \frac{\lambda_k^j}{\rho_k^j}\right)\right]^2.$$
>
> For example, one could add an energy constraint alongside a validity constraint to jointly control both properties during fine-tuning.
> We will add this discussion in the Apx.
>
> **On comparison to Khalafi et al. (2024) [4] and constrained baselines (Q3).**
> Khalafi et al. [4] address *constrained generation* under variational constraints without reward optimization, whereas our work addresses *constrained generative optimization* under expectation constraints.
> These are different problem formulations; we have expanded Sec 7 to clarify this distinction.
>
> We have made further efforts to include a baseline from a different but related setting (test-time vs. finetuning). We have now evaluated DiffOpt [5], a test-time scheme that keeps the model weights fixed, on the illustrative example.
> DiffOpt [5] exhibits a higher violation rate ($10$–$21$% vs. $8.5$% for CFO), while showing $44$–$55\times$ higher per-sample cost. These benchmarking results show the strength of CFO in incurring low violation rates while being compute efficient for sampling.
>
> **References.**
>
> [1] Friede et al., dxtb — an efficient and fully differentiable framework for extended tight-binding. The Journal of Chemical Physics, 161(6), 2024.
>
> [2] Zheng et al., DiffusionNFT: Online Diffusion Reinforcement with Forward Process. ICLR 2026.
>
> [3] De Santi et al., Flow Density Control: Generative Optimization Beyond Entropy-Regularized Fine-Tuning. NeurIPS 2025.
>
> [4] Khalafi et al., Constrained diffusion models via dual training. NeurIPS 2024.
>
> [5] Kong et al., Diffusion Models as Constrained Samplers for Optimization with Unknown Constraints. AISTATS 2025.

---

> > ### Author Rebuttal · Reviewer_mW7n · 2026-04-03
> >
> > I thank the authors for the rebuttal. My concerns have been addressed by the rebuttal. I maintain my positive score in favour of acceptance.

---

### Official Review · Reviewer_xtUc · 2026-03-10

**Soundness:** 3
**Presentation:** 2
**Significance:** 2
**Originality:** 2
**Overall Recommendation:** 4
**Confidence:** 3

**Summary:**

This paper studies constrained optimization for generative models, focusing on fine-tuning flow or diffusion models to maximize a target reward while satisfying domain-specific constraints. The authors formulate this problem as constrained generative optimization and propose Constrained Flow Optimization (CFO), which applies an augmented Lagrangian scheme to convert the constrained objective into a sequence of standard fine-tuning problems. At each iteration, the model is updated using an augmented reward that penalizes constraint violations, while the dual variables controlling the penalty are adjusted based on the observed constraint gap. Experiments on toy examples and a molecular design task show that the method can improve the target property while maintaining the constraint, whereas reward-only baselines tend to violate it.

**Compliance With Llm Reviewing Policy:**

Affirmed.

**Final Justification:**

The authors provided an effective rebuttal that addressed my concerns and improved the overall quality of the paper.

**Key Questions For Authors:**

1. The empirical evaluation focuses mainly on examples and an single molecular design task. Could the authors comment on how the method would perform on other generative optimization problems or datasets? Are there plans to evaluate the approach in additional settings?
2. How would the proposed method compare with other recent approaches designed for constraint-aware generative modeling?
3. The algorithm introduces a sequential fine-tuning procedure with adaptive penalty updates. Could the authors provide more insight into the sensitivity of the method to hyperparameters and the additional computational cost compared to standard reward-guided fine-tuning?

**Limitations:**

yes

**Strengths And Weaknesses:**

Strengths:
1. The paper focuses on a practically relevant setting where generative models need to optimize a target property while respecting explicit constraints, which is common in molecular design and other scientific discovery tasks.
2. The proposed approach can be implemented as a fine-tuning procedure on top of existing flow or diffusion models, without modifying the underlying architecture.
3. Using an augmented Lagrangian scheme to adapt the penalty during training provides a reasonable way to avoid manually tuning constraint weights, which can be unstable in practice.
4. The paper includes both theoretical discussion and empirical results, and the experiments suggest the method can maintain the constraint while still improving the target property in the tested tasks.
Weaknesses:
1. Most of the experiments are either toy examples or a single molecular design task, so it is hard to judge how well the approach would generalize to other generative optimization settings or datasets.
2. The set of baselines is fairly narrow. The paper main compares against reward-only fine-tuning and a manually tuned penalty variant. Including more recent methods that explicitly handle constraints in generative models would help better understand where the proposed approach stands.
3. Some practical aspects are not very clear from the paper. In particular, there is little discussion about sensitivity to hyperparameters or the computational overhead introduced by the sequential fine-tuning procedure, which makes it difficult to evaluate how robust the method would be in practices.

---

> ### Author Rebuttal · Authors · 2026-03-31
>
> We thank the Reviewer for the careful evaluation and for acknowledging the practical relevance of our setting, the flexibility of our approach, and the inclusion of both theoretical and empirical results.
> In the following, we aim to sharply address the concerns raised.
>
> **On experimental scope (W1/Q1).**
> We report in Apx C additional molecular design experiments on a different dataset (QM9) compared to the main text results.
> Furthermore, we show results with a different reward-constraint pair (including a PoseBusters validity constraint), and usage of a differentiable simulator $\texttt{dxTB}$[1] as surrogate function.
> Overall, these experiments constitute a different constrained generative optimization task, which we will better reference within the main text in the revised manuscript.
> As by the submission's title, within this work we center experimental evaluation on molecular design, which is a task of high practical relevance and large community of interest, and for which we believe our experimental evaluation shows promising results.
> Furthermore, as recognized also by Reviewer JXRi (S3), we agree with the Reviewer that CFO is broadly applicable beyond molecular design, although this clearly goes beyond the scope of this work.
>
> Moreover, as presented within the reply to Reviewer JXRi, we have now included DiffusionNFT [3] as a second FineTuningSolver that supports non-differentiable reward and constraint functions.
> The molecular design results using DiffusionNFT broaden CFO's applicability, and potentially other domains, where non-differentiable rewards and/or constraints are of high relevance.
>
> **On baselines and constrained methods (W2 / Q2).**
> Our paper is primarily focused on *constrained generative optimization* (see Sec 3), i.e., reward-guided adaptation under constraint satisfaction: a distinct problem from *constrained generation* (sampling under constraints without reward optimization).
> In particular, our work tackles the *fine-tuning* setting, where one wishes to permanently adapt a pretrained model so that per-sample generation cost remains as low as for the base model.
> To the best of our knowledge, no existing work addresses this problem formulation, and our two baselines approximate it via widely adopted approaches.
>
> - A **fixed-$\mu$ Lagrangian baseline** (Sec 6, Fig 5), spanning 13 orders of magnitude (acknowledged as a strength by Reviewer JXRi, S4).
> - A **rejection sampling baseline** (Apx B), achieving only $40.6$% constraint satisfaction vs. $61.4$% for CFO. We will highlight this benchmark more strongly in the main text.
>
> Based on the Reviewers' concern, we further extended our experimental evaluation and compared CFO against DiffOpt[2], a recent method for constrained generative optimization via inference-time adaptation.
> This is a fundamentally different setting: inference-time methods incur additional cost at every generation step ($44$–$55\times$ slower per sample in our experiments), which is often prohibitive for large-scale sampling campaigns (e.g., virtual screening in molecular design).
> Moreover, since DiffOpt does not leverage off-the-shelf fine-tuning solvers alike CFO, we found it significantly harder to tune in practice.
>
> The results for DiffOpt on the illustrative experiment (Sec 6, Fig 2a-d) are:
>
> | | Reward $\uparrow$ | Violations $\downarrow$ | Sampling time |
> |---|---|---|---|
> | CFO (fine-tuned) | $-4.75$ | $8.5\%$ | $1\times$ |
> | DiffOpt (high reward) | $-4.52$ | $21.4\%$ | $44\times$ |
> | DiffOpt (gentle+long) | $-5.05$ | $10.4\%$ | $55\times$ |
>
> CFO shows the lowest constraint violation rate and comparable reward while being much cheaper to sample.
>
> **On hyperparameter sensitivity and computational cost (W3 / Q3).**
> Extensive ablations over all major hyperparameters ($\rho_{\mathrm{init}}, \eta, \tau, \lambda_{\min}$) are already provided in Apx D.
> These ablations demonstrate that CFO is qualitatively robust, with performance remaining stable across several parameters.
> Regarding computational cost, we already report its analysis in the last paragraph of Sec 6: CFO ($K=6$, $N=10$) requires $44.5$ min versus $40.25$ min for Adjoint Matching ($N=60$), an overhead of approximately $10$%.
> Both methods perform the same number of Adjoint Matching steps ($N=60$); the overhead arises solely from the extra sampling and constraint evaluation in Step 3 of Algorithm 1.
> We will highlight these analyses more prominently in the revised manuscript.
>
> We hope that these further clarifications, especially regarding baselines, ablations, and computational cost, can help the Reviewer further appreciate our work.
>
> **References.**
>
> [1] Friede et al. dxtb — an efficient and fully differentiable framework for extended tight-binding. The Journal of Chemical Physics, 161(6), 2024.
>
> [2] Kong et al., Diffusion Models as Constrained Samplers for Optimization with Unknown Constraints. AISTATS 2025.
>
> [3] Zheng et al., DiffusionNFT: Online Diffusion Reinforcement with Forward Process. ICLR 2026.

---

> > ### Author Rebuttal · Reviewer_xtUc · 2026-04-03
> >
> > The authors’ rebuttal has clarified my concerns, and I will revise my score from 3 to 4.

---

### Official Review · Reviewer_z6ME · 2026-03-11

**Soundness:** 4
**Presentation:** 3
**Significance:** 3
**Originality:** 4
**Overall Recommendation:** 4
**Confidence:** 3

**Summary:**

This paper proposes a constrained control framework for fine-tuning generative flow models.
The key idea is to formulate constrained generative optimization and solve it using an augmented Lagrangian scheme.
Instead of manually tuning a penalty coefficient to balance reward maximization and constraint satisfaction, the method introduces dual variables and quadratic penalties, which are updated iteratively during fine-tuning.
This transforms the constrained optimization problem into a sequence of KL-regularized fine-tuning subproblems, enabling the use of existing generative model training procedures.
Empirically, the method demonstrates improved constraint satisfaction compared to existing reward-guided RL approaches that rely on coefficient sweeps to adjust the trade-off between reward and constraint violations.

**Compliance With Llm Reviewing Policy:**

Affirmed.

**Final Justification:**

This paper presents a principled method for constrained control of generative models. I raised concerns about high dependency from existing solvers and the possibility of mode collapse. The authors provided additional experiments that resolved my concerns, so I am maintaining my positive score.

**Key Questions For Authors:**

1. Can this algorithm be extended to discrete spaces? (e.g., molecular graph optimization)

2. Did the authors try other types of fine-tuning methods besides Adjoint Matching by Domingo-Enrich et al., 2024?

**Limitations:**

None.

**Strengths And Weaknesses:**

Strengths:

1. This paper proposes a principled algorithm with theoretical grounds and a practical form of a method that can be utilized on modern generative models.

2. The motivation is clear, as constrained control problems in generative models are critical.

Weaknesses:

1. This method heavily relies on an existing fine-tuning solver, but it is questionable how it behaves when such a solver is not well-behaved. For example, if the solver leads to mode collapse, the whole process could become problematic. More empirical analysis with many other candidate fine-tuning solvers is required.

---

> ### Author Rebuttal · Authors · 2026-03-31
>
> We thank the Reviewer for regarding the problem tackled as critical, the algorithm principled, practical, and theoretically grounded. In the following, we aim to sharply address the concerns raised.
>
> **On solver dependence and mode collapse (W1).**
> Developing better fine-tuning solvers is an orthogonal and very active area of research.
> Importantly, CFO is solver-agnostic by design, allowing for the leverage of better fine-tuning solvers developed over the next few years.
> Additionally, CFO's sequential structure with multiple outer iterations naturally limits drift per step, as each iteration performs only a moderate update from the current policy.
> Empirically, we observe no mode collapse in any of our experiments (both illustrative and molecular design).
> Nonetheless, to further assess CFO, we include results for DiffusionNFT [1] as a second FineTuningSolver alongside Adjoint Matching. Crucially, DiffusionNFT [1] is a gradient-free method, allowing to apply CFO to non-differentiable rewards and constraints.
> On the illustrative example (Sec 6, Fig 2a-d) and the molecular design task (Sec 6, Fig 3), we report:
>
> | | Illustrative Reward $\uparrow$ | Illustrative Constr. $\downarrow$ | fulfilled | Molecular Reward $\uparrow$ | Molecular Constr. $\downarrow$ | fulfilled |
> |---|---|---|---|---|---|---|
> | PRE | $−7.62$ | $0.58$ | ✗ | $6.55$ | $−77.86$ | ✗ |
> | AM | $-2.93$ | $2.47$ | ✗ | $8.37$ | $-78.31$ | ✗ |
> | CFO (w/ AM) | $-4.75$ | $0.12$ | ✓ | $8.39$ | $-82.28$ | ✓ |
> | NFT | $-3.59$ | $1.76$ | ✗ | $7.92$ | $-77.75$ | ✗ |
> | CFO (w/ NFT) | $-5.28$ | $0.06$ | ✓ | $7.62$ | $-81.05$ | ✓ |
>
> CFO achieves strong constraint satisfaction with both solvers, confirming the modularity of our framework and proving the ease of adaptation to arbitrary, application-specific fine-tuning solvers.
>
> **On extension to discrete spaces (Q1).**
> The CFO scheme is agnostic to the underlying data type.
> In fact, the data modality is handled by the specific FineTuningSolver used.
> Our current experiments already involve optimization over discrete atom types, bond types, and formal charges, alongside continuous coordinates.
> For fully discrete generative models, one can replace Adjoint Matching with a discrete-space solver such as DRAKES [3] or SEPO [4]. Use of CFO in such settings (e.g., for peptide or protein design) is straightforward and a natural direction for future work.
>
> **On other fine-tuning methods (Q2).**
> Yes, we have now added DiffusionNFT [1] as a second FineTuningSolver alongside Adjoint Matching, as reported above (W1).
> DiffusionNFT [1] operates via online policy gradient estimation without requiring differentiable objectives, demonstrating that CFO can leverage fundamentally different optimization strategies and extend to non-differentiable rewards and constraints.
> This extension confirms the solver-agnostic design of the augmented Lagrangian outer loop.
> We will include these results and discussion in the updated manuscript.
>
> **References.**
>
> [1] Zheng et al., DiffusionNFT: Online Diffusion Reinforcement with Forward Process. ICLR 2026.
>
> [2] Venkatraman et al. Amortizing intractable inference in diffusion models for vision, language, and control. NeurIPS 2024.
>
> [3] Wang et al. Fine-Tuning Discrete Diffusion Models via Reward Optimization with Applications to DNA and Protein Design. ICLR 2025.
>
> [4] Oussama Zekri et al. Fine-Tuning Discrete Diffusion Models with Policy Gradient Methods. NeurIPS 2025.

---

> > ### Author Rebuttal · Reviewer_z6ME · 2026-04-01
> >
> > My concerns are resolved; I remain my score positive.

---

### Official Review · Reviewer_JXRi · 2026-03-11

**Soundness:** 3
**Presentation:** 4
**Significance:** 3
**Originality:** 3
**Overall Recommendation:** 6
**Confidence:** 3

**Summary:**

The paper tackles the problem of generating molecules with constraints using flow-based models, specifically through the finetuning of an existing pretrained generative flow model which may not be aware of the specific constraints desired for a task. Existing works include constraints as penalties in the objective, which requires the (unreliable and costly) tuning of the penalty weights. The authors propose the CFO framework, which performs a series of steps to optimize the objective, including updating the penalty terms to improve constraint satisfaction.

**Compliance With Llm Reviewing Policy:**

Affirmed.

**Final Justification:**

The authors have significantly strengthened the paper throughout the rebuttal process. I like the core idea, it is presented well, and the experimental evaluation is solid. As a result, I raised my score to be strongly in favour of acceptance.

**Key Questions For Authors:**

1. Do you see a way of making constraint satisfaction exact with your method? In cases where constraints are safety-critical, this can be an important factor whether a system can be used or not.
2. What do you think are the main challenges of applying it on domains outside of molecules? Is it the need for a differentiable reward and constraint calculator, at least when CFO is used with Adjoint Matching? How realistic is this differentiability requirement? Discussing this would perhaps clear up some doubts I have about its significance in other domains.
3. Do you see a way of handling discrete constraints, such as "I want rings in my molecule to have at most 6 atoms", which is a reasonable constraint for obtaining stable molecules. Addressing this would improve the generality of the types of constraints that can be handled.

**Limitations:**

yes

**Strengths And Weaknesses:**

# Strengths
- Excellent framing of the problem, its importance, and how existing solutions fall short [Presentation+]
- Nuanced discussion of experimental results [Presentation+]
- While CFO is primarily applied to the molecular domain here, it seems easily applicable to other domains as well. [Significance+]
- The CFO framework has clear benefits over choosing a fixed $\mu$ in theory and practice [Soundness+]
- Thorough experiments, including ablations of key hyperparams in the appendix [Soundness+]
- The CFO framework is a clever combination of various existing ideas [Originality+]

# Weaknesses
1. In practice, the constraints are still violated to some degree. While the authors provide a potential explanation, the hypothesis is not tested. [Soundness-]
2. Given the importance of the choice of FineTuningSolver in terms of the theoretical conditions, it would have been nice to see at least one other alternative to the Adjoint Matching used to see the dependence of CFO on the quality of the choice of FineTuningSolver. [Soundness-]
3. Due to point 2, while the requirement of differentiability of reward and constraint functions is indeed not due to CFO but due to the FineTuningSolver used, the authors do not mention any concrete alternatives to Adjoint Matching. So, it is not clear to me whether this requirement of differentiability can be reasonably avoided in practice, especially given Theorem 5.4 requiring good performance by the FineTuningSolver. [Significance-]
4. Section 4 states what the steps are and what they accomplish, but it is not particularly clear to me why these exact steps are used, in that order. I recognize that these are related to the AL scheme, but a clearer exposition and connection to AL would make this section easier to follow. [Presentation-]

---

> ### Author Rebuttal · Authors · 2026-03-31
>
> We thank the Reviewer for regarding our problem framing as excellent, and the experiments as thorough and showing clear benefits.
>
> **On residual constraint violations (W1).**
> We appreciate the Reviewer's attention to this point.
> This phenomenon is discussed in Sec 6 and further analyzed in Apx D (Table 4).
> Our analysis shows that the remaining violations are primarily due to the finite number of samples used to estimate expectations and gradients during fine-tuning: we show that increasing the sample count reduces violations (Apx D, Table 4).
> Minor residual effects from the stochastic optimization procedure may also contribute.
>
> **On alternative FineTuningSolvers (W2).**
> We agree that demonstrating CFO with a second solver strengthens the modularity claim.
> Thus we have now **included DiffusionNFT [1] as a second FineTuningSolver.**
> We report results on both the illustrative example (Sec 6, Fig 2) and the molecular design task (Sec 6, Fig 3, constraint energy below $-80$ Ha)):
>
> | | Illustrative Reward $\uparrow$ | Illustrative Constr. $\downarrow$ | fulfilled | Molecular Reward $\uparrow$ | Molecular Constr. $\downarrow$ | fulfilled |
> |---|---|---|---|---|---|---|
> | PRE | $−7.62$ | $0.58$ | ✗ | $6.55$ | $−77.86$ | ✗ |
> | AM | $-2.93$ | $2.47$ | ✗ | $8.37$ | $-78.31$ | ✗ |
> | CFO (w/ AM) | $-4.75$ | $0.12$ | ✓ | $8.39$ | $-82.28$ | ✓ |
> | NFT | $-3.59$ | $1.76$ | ✗ | $7.92$ | $-77.75$ | ✗ |
> | CFO (w/ NFT) | $-5.28$ | $0.06$ | ✓ | $7.62$ | $-81.05$ | ✓ |
>
> CFO achieves strong constraint satisfaction with both solvers on both tasks, confirming that CFO is solver-agnostic in practice.
>
> **On differentiability (W3).**
> Our newly added implementation using DiffusionNFT [1], a recent yet widely adopted FineTuningSolver (see W2), does not require any gradients, allowing broad applicability of CFO across domains.
> Since DiffusionNFT [1], alike alternative gradient-free methods such as FlowGRPO [2], show strong performances on several application domains, they can straightforwardly be employed within CFO.
> We will include and discuss this extension to gradient-free oracles within the updated manuscript.
>
> **On Sec 4 presentation (W4).**
> We thank the Reviewer for this suggestion.
> The idea behind CFO is to introduce two dual variables: $\rho$, which penalizes constraint violations via a quadratic term, and $\lambda$, which shifts the penalty to concentrate around the constraint boundary.
> These together define an augmented reward (Step 1).
> Given this augmented reward, one fine-tunes the model using any off-the-shelf solver (Step 2).
> The remaining steps evaluate how well the updated model satisfies the constraint (Step 3) and adapt the dual variables for the next round (Steps 4–5).
> We will revise Sec 4 to make this connection to dual optimization schemes more explicit.
>
> **On exact constraint satisfaction (Q1).**
> CFO optimizes an expectation constraint $\mathbb{E}[c(x)] \leq B$ and provides guarantees at the distributional level.
> Reformulating the constraint as $\mathbb{E}[\mathrm{ReLU}(c(x) - B)] \leq 0$ could directly penalize each individual violation and provide a stronger per-sample signal.
> We acknowledge that absolute per-sample feasibility is a fundamentally different problem; we view this as a natural extension.
> For safety-critical applications, post-filtering generated samples provides a practical complementary strategy.
> CFO makes post-filtering more effective: on the molecular design task, CFO achieves $61.4$% per-sample satisfaction vs. $40.6$% for AM, yielding more valid candidates at lower rejection cost.
>
> **On applicability outside molecules (Q2).**
> Differentiable oracles are readily available in many domains spanning biology and image generation (e.g., AlphaFold [3] confidence, CLIPScore [4] for text-to-image).
> For black-box settings (e.g., physics simulators), our newly added implementation with DiffusionNFT [1] can be leveraged.
> We do not see a fundamental challenge in applying CFO to new domains.
>
> **On discrete constraints (Q3).**
> In Apx C, we already show results with a learned PoseBusters validity constraint, which is itself a binary (discrete) constraint function.
> More generally, a constraint such as ring size (6 atoms per ring) can be encoded via an oracle that returns $0$ when satisfied and $1$ otherwise, with $B = 0$, such that $\mathbb{E}[c(x)] \leq 0$ enforces the constraint on average.
> With the inclusion of DiffusionNFT [1], which supports non-differentiable objectives, discrete and binary constraints become directly accessible within CFO.
>
> **References.**
>
> [1] Zheng et al., DiffusionNFT: Online Diffusion Reinforcement with Forward Process. ICLR 2026.
>
> [2] Jie Liu et al., Flow-GRPO: Training Flow Matching Models via Online RL. ICLR 2026.
>
> [3] Abramson et al., Accurate structure prediction of biomolecular interactions with AlphaFold 3. Nature 2024.
>
> [4] Hessel et al., CLIPScore: A Reference-free Evaluation Metric for Image Captioning. EMNLP 2021.

---

> > ### Author Rebuttal · Reviewer_JXRi · 2026-03-31
> >
> > I have read all reviews and rebuttals. All my concerns have been sufficiently addressed and I didn't see any other critical weaknesses raised; especially the additional evaluation with a different solver that has fewer requirements is appreciated.
> >
> > I think that this is a very strong submission and happily recommend acceptance.

---

### Decision · Program_Chairs · 2026-04-30

**Decision:**

Accept (regular)

**Comment:**

This paper addresses the problem of fine-tuning pre-trained flow and diffusion models to maximize a reward function while satisfying domain-specific constraints. The key idea is to apply an augmented Lagrangian scheme that converts the constrained optimization problem into a sequence of standard KL-regularized fine-tuning subproblems.  All four reviewers had positive review of the paper after the rebuttal. I recommend the authors to consider updating the paper in the final version incorporating all comments from the reviewers.